# Coloring the Noise: Adversarial Sobolev Alignment for Faithful Image Super Resolution

Hongbo Wang [† 1 2 3]  Huaibo Huang [1 2]  Pin Wang [1 2]  Jinhua Hao [3]  Chao Zhou [3]  Ran He [1 2]

Code: https://github.com/wafer-bob/ASASR

## Abstract

Generative priors in Image Super-Resolution (SR) often compromise faithful restoration, we attribute this limitation to a fundamental spectral misalignment between isotropic objectives and the intrinsic natural image manifold. While Direct Preference Optimization offers a path to alignment, its reliance on spectrally flat Gaussian noise fails to distinguish authentic high-frequency details from hallucinations. To bridge this geometric gap, we propose **ASASR**, a theoretically grounded framework that recasts the generative flow into a Sobolev-induced Riemannian geometry by explicitly coloring the noise transition kernel to mirror natural spectral decay. Driving this geometric alignment, we integrate a parametric adversary grounded in the Riesz Representation Theorem, which synthesizes targeted negative samples equivalent to worst-case Sobolev gradients to direct optimization along the tangent space of plausible structural failures. Extensive evaluations demonstrate that ASASR outperforms leading generative baselines, particularly in preserving spectral consistency and structural fidelity, offering a robust solution that effectively mitigates artifacts.

## 1. Introduction

Image Super-Resolution (SR) functions as a fundamental inverse problem, aiming to reconstruct high-quality (HQ) images from their degraded low-quality (LQ) counterparts.

Leveraging the powerful generative priors of large-scale vision models (Saharia et al., 2022; Rombach et al., 2022; Esser et al., 2024; Labs, 2024), recent approaches have achieved significant breakthroughs in synthesizing realistic textures (Wu et al., 2024; Yu et al., 2024). However, the prevailing supervised training paradigm imposes a fundamental ceiling on faithful restoration, as it inherently anchors the optimization process to synthetic degradation priors rather than the authentic natural image manifold. Given that real-world degradation is often unknown and ill-posed, standard methods act by enforcing strict pixel-wise alignment on synthetic data pairs. This rigid dependency compels the model to overfit to artificial degradation assumptions, effectively prioritizing the memorization of synthetic patterns over the capture of authentic natural textures.

To explicitly penalize such stochastic deviations and enforce manifold adherence, adapting Direct Preference Optimization (DPO) (Rafailov et al., 2024) offers a potential path alignment. However, we attribute the limited efficacy of standard DPO in SR to a fundamental geometric flaw: its reliance on naive isotropic Gaussian parameterization (Ho et al., 2020). Specifically, as shown in Fig. 3, our analysis reveals that this spectrally flat prior diverges sharply from the intrinsic spectral decay of natural images (Field, 1987; Rahaman et al., 2019). While acceptable for Text-to-Image (T2I) synthesis, this spectral misalignment proves detrimental for Super-Resolution, which demands strict high-frequency fidelity. Consequently, lacking the inductive bias to differentiate authentic details from spurious noise, such isotropic objectives inevitably yield high-frequency artifacts that violate the data manifold.

In response to these challenges, we propose **ASASR**: **A**dversarial **S**obolev **A**lignment for **S**uper-**R**esolution, a theoretically grounded framework that induces natural manifold constraints for faithful image super-resolution. Specifically, we introduce *Sobolev Spectral Rectification (SSR)* to color the noise in the data representation, parameterizing the transition kernel via Colored Gaussian Noise defined by a structured covariance matrix that explicitly mirrors the spectral density of natural textures. Crucially, we derive that this alignment with natural statistics fundamentally re-

†Work was done during an internship at Kuaishou. [1]MAIS & NLPR, Institute of Automation, Chinese Academy of Sciences, Beijing, China [2]School of Artificial Intelligence, University of Chinese Academy of Sciences, Beijing, China [3]Kuaishou Technology, Beijing, China. Correspondence to: Huaibo Huang <huaibo.huang@cripac.ia.ac.cn>.

*Proceedings of the 43$^{rd}$ International Conference on Machine Learning*, Seoul, South Korea. PMLR 306, 2026. Copyright 2026 by the author(s).

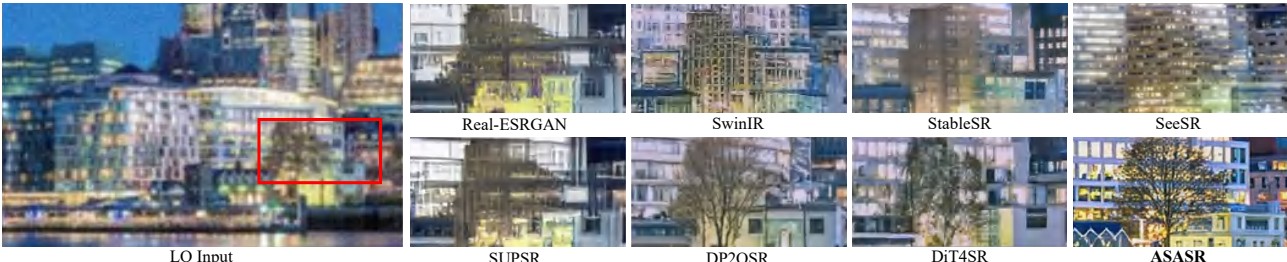

*Figure 1.* Visual comparison with state-of-the-art SR methods. The proposed ASASR achieves superior perceptual quality, generating more realistic textures and faithful structural details from the low-quality input.

shapes the optimization objective, mathematically evolving the implicit distance metric into the Sobolev norm $H^s$. By traversing the solution space within this Sobolev-induced Riemannian geometry, the model gains a frequency-aware inductive bias, enabling the precise rectification of structural artifacts that are otherwise invisible to isotropic priors.

However, the geometric precision established by SSR remains dormant without challenging supervisory signals to drive it. A critical bottleneck in standard DPO paradigms stems from the scarcity of informative negative samples, where static pairs often fail to capture the subtle structural nuances required for the ill-posed nature of super-resolution. To transcend this limitation, we introduce *Adversarial Manifold Guidance (AMG)*, a parametric adversary that characterizes the manifold of realistic artifacts to synthesize targeted, semantically aligned negatives on the fly. Integrating this dynamic supervision establishes our Adversarial Sobolev DPO (AS-DPO) framework. Grounded in the Riesz Representation Theorem, we demonstrate that AS-DPO leverages these perturbations as worst-case Sobolev gradients, steering optimization along the tangent space of plausible perceptual failures.

To empirically validate our framework, we conduct extensive evaluations on super-resolution benchmarks and downstream high-level vision tasks, benchmarking ASASR against leading diffusion-based and GAN-based approaches. Beyond conventional distortion and perceptual metrics, we also evaluate spectral fidelity to assess how closely the generated frequency profiles match the ground truth. Our results demonstrate that ASASR achieves superior performance in balancing fidelity and realism and boosting downstream accuracy, successfully reconstructing high-frequency textures that align with natural image statistics while effectively mitigating artifacts.

Our contributions can be summarized as follows:

- We identify the spectral disparity between isotropic generative priors and natural image statistics as a critical bottleneck in SR, and propose ASASR, a geometry-aware framework designed to bridge this gap by en-

forcing strict manifold adherence.

- We introduce Sobolev Spectral Rectification to color transport noise, reshaping the implicit optimization metric to align with natural spectral statistics and rectify frequency-biased hallucinations.

- We propose Adversarial Manifold Guidance to synthesize targeted negatives, proving that these perturbations constitute worst-case Sobolev gradients specifically driving optimization along the tangent space of plausible failures.

- Extensive evaluations demonstrate that ASASR achieves superior performance over leading generative methods, particularly in preserving spectral consistency and fine-grained structural fidelity.

## 2. Background

**Flow Matching.** We construct our approach within the Flow Matching framework (Lipman et al., 2023), adapted here for conditional generation. Let $\boldsymbol{x}_1 \sim q(\boldsymbol{x}_1|\boldsymbol{c})$ denote the high-resolution data distribution conditioned on the low-resolution input $\boldsymbol{c}$, and let $\boldsymbol{x}_0 \sim p(\boldsymbol{x}_0) = \mathcal{N}(\boldsymbol{0}, \mathbf{I})$ be the prior noise distribution. We define a conditional probability path between noise and data via linear interpolation:

$$\boldsymbol{x}_t = (1-t)\boldsymbol{x}_0 + t\boldsymbol{x}_1, \quad t \in [0,1], \quad (1)$$

Differentiating with respect to time yields the conditional vector field $\boldsymbol{u}_t(\boldsymbol{x}|\boldsymbol{x}_0, \boldsymbol{x}_1) = \boldsymbol{x}_1 - \boldsymbol{x}_0$. To approximate this field, a velocity network $\boldsymbol{v}_\theta(\boldsymbol{x}_t, t, \boldsymbol{c})$ is trained by minimizing the conditional flow matching objective:

$$\mathcal{L}_{\text{CFM}}(\theta) = \mathbb{E}_{t, \boldsymbol{x}_0, \boldsymbol{x}_1} \left[ \|\boldsymbol{v}_\theta(\boldsymbol{x}_t, t, \boldsymbol{c}) - (\boldsymbol{x}_1 - \boldsymbol{x}_0)\|^2 \right], \quad (2)$$

**Direct Preference Optimization.** To align the generative prior with human perception, we employ DPO (Rafailov et al., 2024) using preference triplets $\mathcal{D} = \{(\boldsymbol{c}, \boldsymbol{x}_1^w, \boldsymbol{x}_1^l)\}$. To circumvent intractable likelihood computation, we extend the objective to the trajectory space governed by ODE.

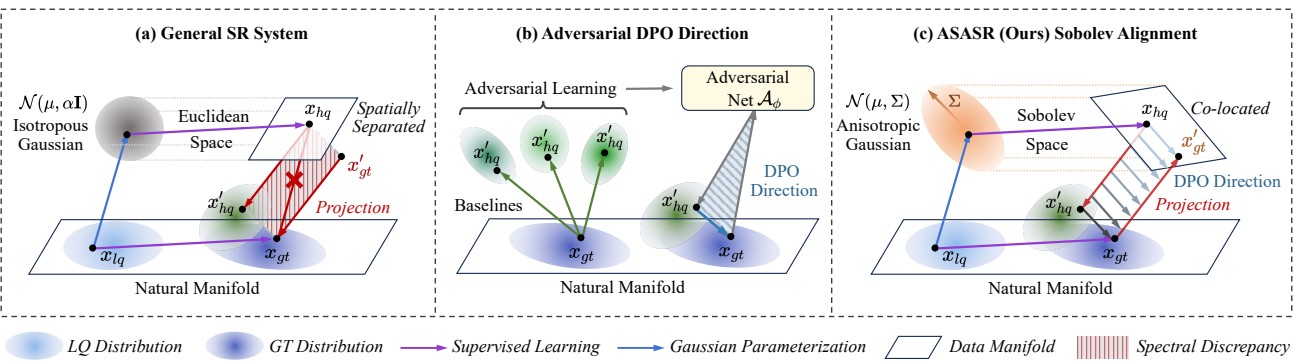

*Figure 2.* Conceptual illustration of spectral misalignment and our proposed ASASR. **(a)** Standard SR frameworks always assume an isotropic Euclidean space, a simplification that neglects the intrinsic spectral nature of real-world data. This geometric mismatch projects the generated candidate $\boldsymbol{x}_{hq}$ onto a manifold disjoint from the ground truth $\boldsymbol{x}_{gt}$, resulting in the significant spectral discrepancy (hatched region). **(b)** To bridge this geometric gap, we characterize the manifold of realistic artifacts using diverse baseline hypotheses (green nodes). The adversary $\mathcal{A}_\phi$ approximates the distribution of these plausible inconsistencies by learning the mapping from $\boldsymbol{x}_{gt}$, effectively encoding generative artifact patterns into a learnable prior. **(c)** Our ASASR framework integrates this targeted guidance within a reshaped Sobolev geometry. By enforcing an anisotropic Gaussian prior, we rectify the optimization landscape, ensuring the projection trajectory is spectrally accurate and co-located with the target distribution, effectively resolving the misalignment in (a).

By optimizing over the resulting deterministic evolution paths $\boldsymbol{x}_{0:1}$, the trajectory-wise loss is defined as:

$$
\begin{aligned}
\mathcal{L}_{\text{DPO}}(\theta) = -\mathbb{E}_{(\boldsymbol{c},\boldsymbol{x}_1^w,\boldsymbol{x}_1^l)\sim\mathcal{D}}&\Bigg[\log\sigma\Bigg(\\
\beta\bigg[\log\frac{p_\theta(\boldsymbol{x}_{0:1}^w|\boldsymbol{c})}{p_{\text{ref}}(\boldsymbol{x}_{0:1}^w|\boldsymbol{c})} &- \log\frac{p_\theta(\boldsymbol{x}_{0:1}^l|\boldsymbol{c})}{p_{\text{ref}}(\boldsymbol{x}_{0:1}^l|\boldsymbol{c})}\bigg]\Bigg)\Bigg],
\end{aligned}
\tag{3}
$$

where $\sigma(\cdot)$ denotes the logistic sigmoid function, the temperature hyperparameter $\beta$ controls regularization.

**Sobolev Space.** The Sobolev space $H^s(\Omega)$ (Adam & Fournier, 2003) provides a rigorous basis for quantifying regularity on $\Omega \subset \mathbb{R}^2$. Unlike the standard $L^2$ framework that treats pixel intensity independently, $H^s(\Omega)$ explicitly incorporates smoothness constraints. Ideally viewed through a spectral lens, $H^s$ is defined by a strict decay condition on Fourier coefficients, ensuring that signal energy is predominantly concentrated in low-frequency modes. Formally, assuming periodic boundaries with frequency vector $\boldsymbol{\omega} \in \mathbb{Z}^2$ and coefficients $\hat{f}(\boldsymbol{\omega})$, the space is defined for $s \geq 0$ as:

$$
H^s(\Omega) = \Big\{f \in L^2 : \sum_{\boldsymbol{\omega}}(1+\|\boldsymbol{\omega}\|^2)^s|\hat{f}(\boldsymbol{\omega})|^2 < \infty\Big\}, \tag{4}
$$

where the term $(1+\|\boldsymbol{\omega}\|^2)^s$ acts as a frequency-dependent penalty enforcing natural spectral distribution. This space is equipped with the inner product and norm:

$$
\langle f,g\rangle_{H^s} = \sum_{\boldsymbol{\omega}}(1+\|\boldsymbol{\omega}\|^2)^s\hat{f}(\boldsymbol{\omega})^*\hat{g}(\boldsymbol{\omega}), \tag{5}
$$

$$
\|f\|_{H^s}^2 = \langle f,f\rangle_{H^s}, \tag{6}
$$

## 3. Manifold Rectification in Sobolev Space

### 3.1. The Spectral Deficiency of Euclidean DPO

To make the optimization of Eq. (3) tractable, we approximate the continuous probability flow using standard Euler integration. Following the Flow Matching formulation, we parameterize the local transitions for the ideal posterior $q$ and the policy $p_\psi$ as Gaussian approximations centered on the deterministic ODE trajectories:

$$
q(\boldsymbol{x}_{t-\Delta t}|\boldsymbol{x}_t,\boldsymbol{x}_1) = \mathcal{N}\left(\boldsymbol{x}_t - \frac{\boldsymbol{x}_1-\boldsymbol{x}_t}{1-t}\Delta t, \eta^2\Delta t\mathbf{I}\right), \tag{7}
$$

$$
p_\psi(\boldsymbol{x}_{t-\Delta t}|\boldsymbol{x}_t,\boldsymbol{c}) = \mathcal{N}\left(\boldsymbol{x}_t - \boldsymbol{v}_\psi(\boldsymbol{x}_t,t,\boldsymbol{c})\Delta t, \eta^2\Delta t\mathbf{I}\right), \tag{8}
$$

where $\psi \in \{\theta, \text{ref}\}$, $\eta$ represents an auxiliary variance parameter for the likelihood definition. Crucially, this isotropic parametrization implies an underlying Euclidean geometry. Let $\boldsymbol{\gamma}_\theta$ denote the residual between the model prediction and the target vector field:

$$
\boldsymbol{\gamma}_\theta(\boldsymbol{x}_t,\boldsymbol{c}) := \boldsymbol{v}_\theta(\boldsymbol{x}_t,\boldsymbol{c}) - \boldsymbol{v}_q(\boldsymbol{x}_t), \tag{9}
$$

Consequently, the log-likelihood ratio objective reduces to a difference of squared $\ell_2$ norms:

$$
\log\frac{p_\theta(\boldsymbol{x}_{t-\Delta t}|\boldsymbol{x}_t,\boldsymbol{c})}{p_{\text{ref}}(\boldsymbol{x}_{t-\Delta t}|\boldsymbol{x}_t,\boldsymbol{c})} \propto -\left(\|\boldsymbol{\gamma}_\theta\|_2^2 - \|\boldsymbol{\gamma}_{\text{ref}}\|_2^2\right), \tag{10}
$$

However, we trace the root of the identified spectral misalignment to this reliance on the $\ell_2$ norm. Invoking Parseval's theorem, we can express the spatial error in the frequency domain as:

$$
\|\boldsymbol{\gamma}_\theta\|_2^2 = \|\mathcal{F}(\boldsymbol{\gamma}_\theta)\|_2^2 = \frac{1}{MN}\sum_{\boldsymbol{k}}\underbrace{1}_{\text{weight}}\cdot\|\hat{\boldsymbol{\gamma}}_\theta[\boldsymbol{k}]\|^2, \tag{11}
$$

where $\mathcal{F}$ denotes the Fourier transform, $\hat{\boldsymbol{\gamma}}_\theta[\boldsymbol{k}]$ represents the spectral error component at frequency index $\boldsymbol{k}$, $\hat{\boldsymbol{\gamma}}_\theta(\boldsymbol{\omega})$ represents the spectral error component at frequency $\boldsymbol{\omega}$, $M$ and $N$ denote the spatial dimensions of the image. This identity explicitly demonstrates that the optimization imposes a uniform weighting across the entire spectrum. Such spectral indifference proves catastrophic in practice. As visualized in Fig. 3, the standard $\ell_2$ objective fails to counteract the inherent spectral bias of neural networks (Rahaman et al., 2019), causing the learned distribution to diverge significantly from natural image statistics in the high-frequency region. This spectral deficit is not merely theoretical; it directly manifests as the loss of fine texture and artifacts. Detailed derivation is provided in Appendix A.1.

### 3.2. Reshaping Optimization Geometry via Sobolev Spectral Rectification

To mitigate the frequency-agnostic nature of standard Euclidean objectives, we propose *Sobolev Spectral Rectification*, a method that reshapes the underlying optimization geometry by substituting the isotropic noise assumption with colored Gaussian noise. Formally, we generalize the transition kernels in Eqs. (7)–(8) by replacing the identity covariance $\mathbf{I}$ with a structured spectral operator $\boldsymbol{\Sigma}_s$:

$$\boldsymbol{\Sigma}_s = \mathcal{F}^{-1}\mathrm{diag}\big((1 + \|\boldsymbol{\omega}\|_2^2)^{-s}\big)\,\mathcal{F}, \quad (12)$$

This structured covariance induces a fundamental shift in the optimization metric. As the Gaussian likelihood is governed by the Mahalanobis distance, the learning signal is shaped by the precision matrix $\boldsymbol{\Sigma}_s^{-1}$. While $\boldsymbol{\Sigma}_s$ acts as a low-pass filter, its inverse $\boldsymbol{\Sigma}_s^{-1}$ amplifies high-frequency components, thereby imposing strictly higher penalties on fine-grained discrepancies. Analytically, $\boldsymbol{\Sigma}_s^{-1}$ recovers the Sobolev inner product operator as introduced in Eq. (5), effectively lifting the optimization from flat Euclidean space to the weighted Sobolev manifold $H^s(\Omega)$. Consequently, the log-likelihood ratio in Eq. (24) can be explicitly reformulated as a difference of squared Sobolev norms:

$$\log \frac{p_\theta(\boldsymbol{x}_{t-\Delta t}\,|\,\boldsymbol{x}_t, \boldsymbol{c})}{p_{\mathrm{ref}}(\boldsymbol{x}_{t-\Delta t}\,|\,\boldsymbol{x}_t, \boldsymbol{c})} \propto -\big(\|\boldsymbol{\gamma}_\theta\|_{H^s}^2 - \|\boldsymbol{\gamma}_{\mathrm{ref}}\|_{H^s}^2\big), \quad (13)$$

This formulation offers a compelling physical interpretation: the preference optimization is driven by the spectral energy of the restoration errors conditioned on $\boldsymbol{c}$. We define the Sobolev Energy Gap, $\Delta\mathcal{E}_{H^s}$, to quantify the policy's relative advantage in recovering frequency-weighted details:

$$\Delta\mathcal{E}_{H^s}(\boldsymbol{x}_t, \boldsymbol{c}) \coloneqq \|\boldsymbol{\gamma}_\theta(\boldsymbol{x}_t, \boldsymbol{c})\|_{H^s}^2 - \|\boldsymbol{\gamma}_{\mathrm{ref}}(\boldsymbol{x}_t, \boldsymbol{c})\|_{H^s}^2, \quad (14)$$

Since Gaussian likelihoods imply a reward $r \propto -\|\boldsymbol{\gamma}_\theta\|_{H^s}^2$, maximizing preference is mathematically equivalent to minimizing the energy difference $\Delta\mathcal{E}_{H^s}(\boldsymbol{x}^w) - \Delta\mathcal{E}_{H^s}(\boldsymbol{x}^l)$. Substituting this energy margin into the logistic loss yields our

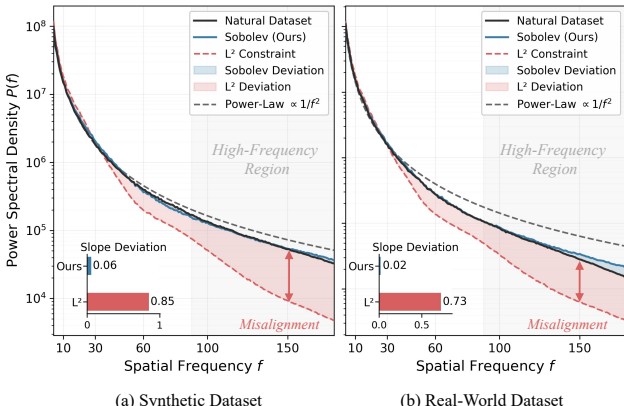

(a) Synthetic Dataset     (b) Real-World Dataset

*Figure 3.* Power Spectral Density analysis relative to the Natural Dataset. The $\ell^2$ baseline exhibits noticeable decay in high frequencies, illustrating the spectral bias inherent to Euclidean constraint. In contrast, our Sobolev constraint closely aligns with the empirical distribution, effectively preserving fine-grained structural fidelity.

final objective, the S-DPO:

$$\begin{aligned}\mathcal{L}_{\text{S-DPO}}(\theta) = -\mathbb{E}_{(\boldsymbol{c}, \boldsymbol{x}_1^w, \boldsymbol{x}_1^l)\sim\mathcal{D}, t\sim\mathcal{U}(0,T)}\Big[\log\sigma\Big(\\ \beta\Big[\Delta\mathcal{E}_{H^s}(\boldsymbol{x}_t^l, \boldsymbol{c}) - \Delta\mathcal{E}_{H^s}(\boldsymbol{x}_t^w, \boldsymbol{c})\Big]\Big)\Big],\end{aligned} \quad (15)$$

Detailed derivation is provided in Appendix A.2.

## 4. Adversarial Manifold Guidance

### 4.1. AS-DPO for Aligned Preference Learning

With the S-DPO objective established, acquiring suitable preference data is critical. Standard SR datasets are incompatible, offering regression-oriented pairs $(\boldsymbol{x}_{\mathrm{LQ}}, \boldsymbol{x}_{\mathrm{GT}})$ rather than comparative triplets. Furthermore, existing T2I preference datasets suffer from weak spatial correspondence: since they rank distinct seeds, differences stem from semantic layout rather than restoration quality (Lu et al., 2025; Zhu et al., 2025). This prevents $\boldsymbol{x}^l$ from serving as a spatially aligned negative, obscuring genuine structural degradation.

In response to these challenges, we introduce *Adversarial Manifold Guidance (AMG)*, a parametric adversary $\mathcal{A}_\phi$ designed to capture the manifold of realistic artifacts for on-the-fly preference synthesis. To train $\mathcal{A}_\phi$, we leverage outputs from standard baselines as proxies to approximate realistic artifacts, optimizing the network to faithfully mimic typical reconstruction failures found in real-world models. With this trained adversary, we employ a coupled sampling strategy to ensure strict semantic alignment. Starting from an intermediate winner state $\boldsymbol{x}_t^w$ along the conditional path (Eq. (1)), the adversary predicts a velocity field $\boldsymbol{v}_\phi$ that steers the trajectory toward a degraded estimate $\hat{\boldsymbol{x}}_1^a$. By forecasting the terminal state via linear extrapolation, we

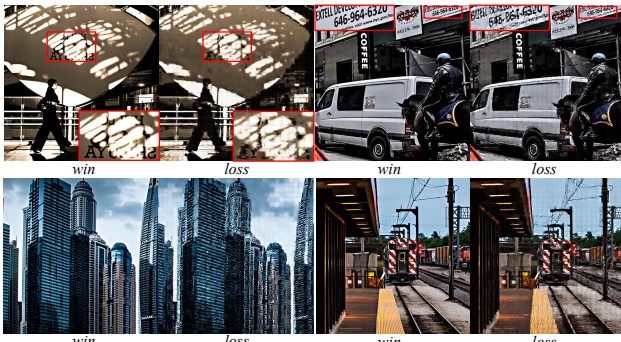

*Figure 4.* Visualization of targeted negatives synthesized. These samples constitute realistic structural artifacts, such as text deformations and architectural distortions, serving as hard negatives.

derive:

$$\widehat{\boldsymbol{x}}_1^a = \boldsymbol{x}_t^w + (1 - t) \cdot \boldsymbol{v}_\phi(\boldsymbol{x}_t^w, t, \boldsymbol{c}), \qquad (16)$$

Critically, to enforce semantic alignment, we re-project this degraded estimate back to the flow state $\boldsymbol{x}_t^a$ using the *identical* noise realization $\boldsymbol{x}_0$ from the winner branch:

$$\boldsymbol{x}_t^a = (1 - t)\boldsymbol{x}_0 + t\widehat{\boldsymbol{x}}_1^a, \qquad (17)$$

By strictly enforcing a shared noise realization $(t, \boldsymbol{\epsilon})$ across both trajectories, the resulting pair $(\boldsymbol{x}_t^w, \boldsymbol{x}_t^a)$ achieves precise semantic alignment. This constraint effectively generates a counterfactual negative sample, isolating perceptual degradations directly tied to the image content rather than random stochastic variations. Integrating this aligned generation strategy into our framework, we reformulate the objective in Eq. (29) into its adversarial variant, termed *AS-DPO*:

$$\mathcal{L}_{\text{AS-DPO}}(\theta) = -\mathbb{E}_{(\boldsymbol{c},\boldsymbol{x}_1^w)\sim\mathcal{D},\, \boldsymbol{x}_t^a\sim\mathcal{A}_\phi,\, t\sim\mathcal{U}(0,T)}\Big[\log$$
$$\sigma\Big(\beta\Big[\Delta\mathcal{E}_{H^s}(\boldsymbol{x}_t^a, \boldsymbol{c}) - \Delta\mathcal{E}_{H^s}(\boldsymbol{x}_t^w, \boldsymbol{c})\Big]\Big)\Big], \qquad (18)$$

### 4.2. Exploiting Model Confidence in Structural Artifacts

While coupled sampling ensures semantic alignment, preference learning efficacy depends on the informativeness of negative samples. A critical pathology in generative models is their tendency to exhibit *Misaligned Confidence*: assigning low residual energy (high likelihood) not only to ground truth data but also to samples containing significant structural degradations. We term these *Spectral Artifacts*: coherent structural hallucinations patterns—such as textural distortions or aliasing—that the model fails to penalize due to the inherent biases of the $\ell_2$ training objective.

To construct informative hard negatives, our AMG targets Misaligned Confidence—a pathology where models assign

high likelihood (low residual) to structurally degraded samples. Unlike standard adversaries that maximize loss to generate noise, $\mathcal{A}_\phi$ seeks to expose these blind spots. Specifically, we search for perturbations that minimize the Euclidean residual energy $\mathcal{J}_{L^2}$—mimicking the model's intrinsic confidence—while forcefully deviating from the ground truth trajectory via a Sobolev constraint.

**Proposition 4.1.** *We formulate the synthesis of the hard negative $\boldsymbol{x}_t^a$ as minimizing the Euclidean residual energy $\mathcal{J}_{L^2}(\boldsymbol{x}) \coloneqq \|\gamma_\theta(\boldsymbol{x})\|_2^2$, subject to a structural trust region defined by the Sobolev norm $\|\boldsymbol{\delta}_t\|_{H^s} \le \varepsilon_t$. The optimal adversarial perturbation $\boldsymbol{\delta}_t^*$ is given by:*

$$\boldsymbol{\delta}_t^* = -\varepsilon_t \frac{\boldsymbol{\Sigma}_s \nabla_{\boldsymbol{x}} \mathcal{J}_{L^2}}{\sqrt{\langle \nabla_{\boldsymbol{x}} \mathcal{J}_{L^2}, \boldsymbol{\Sigma}_s \nabla_{\boldsymbol{x}} \mathcal{J}_{L^2}\rangle_{L^2}}}, \qquad (19)$$

This formulation addresses the core challenge of preference learning: distinguishing between realistic details and model hallucinations. By driving the perturbation along the negative gradient of the $\ell_2$ energy, the adversary actively seeks states where the model is deceptively confident despite the presence of visual degradation. Crucially, the spectral preconditioner $\boldsymbol{\Sigma}_s$ prevents the adversary from collapsing into trivial random noise (which is easily filtered). Instead, it steers the generation towards coherent structural artifacts, structured errors that mimic the model's intrinsic failure modes, thereby providing high-value training signals for S-DPO. Detailed derivation is provided in Appendix B.1.

Since computing Eq. (19) explicitly is intractable, we employ a parametric adversary $\mathcal{A}_\phi$. The following proposition guarantees its theoretical validity.

**Proposition 4.2.** *Let $\phi^*$ be the parameters minimizing the expected energy $\mathbb{E}[\mathcal{J}_{L^2}(\boldsymbol{x}_t + \mathcal{A}_\phi(\boldsymbol{x}_t))]$ subject to $\|\mathcal{A}_\phi\|_{H^s} \le \varepsilon_t$. Assuming sufficient representational capacity, the adversary implicitly recovers the optimal Sobolev direction:*

$$\mathcal{A}_{\phi^*}(\boldsymbol{x}_t) = -\varepsilon_t \frac{\boldsymbol{\Sigma}_s \nabla_{\boldsymbol{x}} \mathcal{J}_{L^2}}{\|\boldsymbol{\Sigma}_s \nabla_{\boldsymbol{x}} \mathcal{J}_{L^2}\|_{H^s}} \equiv \boldsymbol{\delta}_t^*, \qquad (20)$$

where the sufficient representational capacity assumption only means that the parametric adversary can approximate the Sobolev-optimal direction in function space. More details are provided in Appendix B.2. Minimizing the energy loss is locally equivalent to minimizing the inner product $\langle \nabla_{\boldsymbol{x}} \mathcal{J}_{L^2}, \mathcal{A}_\phi\rangle_{L^2}$. Through spectral duality, this forces the network to align anti-parallel to the Sobolev gradient $\nabla_{\boldsymbol{x}}^{H^s} \mathcal{J}_{L^2}$, thereby distilling the optimal spectral descent step into a single forward pass. Detailed derivation is provided in Appendix B.2.

*Table 1.* Quantitative comparison with state-of-the-art real-world SR methods on both synthetic and real-world benchmarks. Best and second best performance are highlighted in **red** and **blue**, respectively.

| Datasets | Metrics | BSRGAN | Real-ESRGAN | SwinIR-GAN | StableSR | DiffBIR | FaithDiff | SeeSR | SUPSR | Dream-Clear | DP2O-SR | DiT4SR | **ASASR** |
|---|---|---|---|---|---|---|---|---|---|---|---|---|---|
| *DIV2K-Val* | PSNR ↑ | 20.36 | **21.11** | 20.43 | 19.93 | 19.85 | 19.82 | 20.46 | 19.66 | 19.79 | 19.60 | 19.01 | **20.60** |
| | SSIM ↑ | 0.5637 | **0.5870** | 0.5678 | 0.5528 | 0.4720 | 0.5167 | 0.5411 | 0.4902 | 0.5137 | 0.5064 | 0.5050 | **0.6171** |
| | LPIPS ↓ | 0.3899 | 0.3147 | 0.3214 | **0.3016** | 0.3748 | 0.3189 | 0.3325 | 0.3721 | 0.3206 | 0.3130 | 0.3299 | **0.2784** |
| | DISTS ↓ | 0.2410 | 0.2191 | 0.2181 | 0.2040 | 0.2273 | 0.2083 | **0.1986** | 0.2328 | 0.2076 | 0.2042 | 0.2087 | **0.2038** |
| | MANIQA ↑ | 0.3097 | 0.3726 | 0.3505 | 0.4224 | 0.5252 | 0.4349 | 0.5187 | 0.5354 | 0.4878 | **0.5810** | 0.4507 | **0.6519** |
| | MUSIQ ↑ | 54.51 | 61.24 | 58.70 | 66.30 | 66.57 | 70.59 | **70.59** | 64.65 | 60.66 | 70.58 | 68.10 | **71.40** |
| | CLIPIQA+ ↑ | 0.5641 | 0.6126 | 0.5883 | 0.6702 | 0.6897 | 0.6767 | **0.7222** | 0.6753 | 0.6356 | 0.7142 | 0.6991 | **0.7521** |
| *LSDIR-Val* | PSNR ↑ | 16.88 | **17.12** | 16.87 | 16.54 | 16.80 | 16.71 | **16.91** | 16.36 | 16.19 | 16.55 | 15.91 | 16.83 |
| | SSIM ↑ | 0.4413 | **0.4528** | **0.4464** | 0.4012 | 0.3909 | 0.4093 | 0.4208 | 0.3710 | 0.4010 | 0.4049 | 0.3887 | 0.4448 |
| | LPIPS ↓ | 0.3992 | 0.3453 | 0.3572 | 0.3285 | 0.3819 | 0.3239 | 0.3366 | 0.3884 | 0.3335 | **0.3153** | 0.3561 | **0.2854** |
| | DISTS ↓ | 0.2410 | 0.2235 | 0.2209 | 0.2047 | 0.2123 | 0.1956 | 0.1917 | 0.2247 | 0.1942 | **0.1876** | 0.1975 | **0.1757** |
| | MANIQA ↑ | 0.3549 | 0.4271 | 0.3874 | 0.4736 | 0.5678 | 0.4435 | 0.5546 | **0.6392** | 0.6203 | 0.6299 | 0.4988 | **0.6372** |
| | MUSIQ ↑ | 63.09 | 67.53 | 65.12 | 70.80 | 70.26 | 71.83 | **73.26** | 69.18 | 68.01 | 73.11 | 71.69 | **73.97** |
| | CLIPIQA+ ↑ | 0.6184 | 0.6756 | 0.6571 | 0.7089 | 0.7375 | 0.7095 | 0.7451 | 0.7420 | 0.7220 | **0.7773** | 0.7106 | **0.7761** |
| *RealSR* | PSNR ↑ | 23.32 | 24.01 | **24.44** | 23.06 | 23.75 | 23.58 | 23.51 | 23.35 | 23.64 | 23.35 | 21.78 | **24.09** |
| | SSIM ↑ | 0.7301 | **0.7340** | **0.7461** | 0.6772 | 0.6173 | 0.6757 | 0.6944 | 0.6342 | 0.6845 | 0.6602 | 0.6292 | 0.7221 |
| | LPIPS ↓ | 0.2788 | 0.2733 | **0.2535** | 0.3047 | 0.3713 | 0.2895 | 0.3001 | 0.3706 | 0.3225 | 0.3159 | 0.3188 | **0.2497** |
| | DISTS ↓ | 0.2248 | 0.2069 | **0.1938** | 0.2152 | 0.2424 | 0.2108 | 0.2214 | 0.2482 | 0.2417 | 0.2221 | 0.2234 | **0.1986** |
| | MANIQA ↑ | 0.4001 | 0.3764 | 0.3476 | 0.4334 | 0.5047 | 0.4666 | 0.5400 | 0.4502 | 0.4232 | **0.5849** | 0.4602 | **0.5730** |
| | MUSIQ ↑ | 63.83 | 60.43 | 58.76 | 65.44 | 65.15 | 68.65 | **69.12** | 59.19 | 57.50 | 69.09 | 67.94 | **71.78** |
| | CLIPIQA+ ↑ | 0.5939 | 0.5878 | 0.5532 | 0.6506 | 0.6725 | 0.6529 | 0.6915 | 0.6131 | 0.5837 | **0.7134** | 0.6654 | **0.7200** |
| *DrealSR* | PSNR ↑ | 25.13 | **26.28** | 26.22 | 25.82 | 24.81 | 25.13 | 25.96 | 24.77 | **26.24** | 25.46 | 23.72 | 25.77 |
| | SSIM ↑ | 0.7583 | **0.7771** | **0.7753** | 0.7177 | 0.5893 | 0.6736 | 0.7411 | 0.6249 | 0.7189 | 0.6845 | 0.6439 | 0.7356 |
| | LPIPS ↓ | 0.3038 | 0.2880 | **0.2737** | 0.3276 | 0.4806 | 0.3597 | 0.3163 | 0.4347 | 0.3555 | 0.3555 | 0.3700 | **0.2874** |
| | DISTS ↓ | 0.2292 | 0.2087 | **0.2037** | 0.2277 | 0.2962 | 0.2409 | 0.2295 | 0.2767 | 0.3558 | 0.2414 | 0.2444 | **0.2023** |
| | MANIQA ↑ | 0.3587 | 0.3434 | 0.3309 | 0.3901 | 0.4981 | 0.4584 | 0.5047 | 0.4158 | 0.3151 | **0.5641** | 0.4435 | **0.6067** |
| | MUSIQ ↑ | 59.25 | 54.28 | 52.77 | 59.18 | 59.81 | 66.21 | 64.72 | 54.52 | 42.82 | **68.71** | 64.69 | **68.36** |
| | CLIPIQA+ ↑ | 0.5935 | 0.5540 | 0.5355 | 0.6193 | 0.6518 | 0.6604 | 0.6716 | 0.5888 | 0.5176 | **0.6771** | 0.6674 | **0.6803** |

*Table 2.* Quantitative comparison on downstream tasks: OCR, Object Detection, Instance Segmentation, and Semantic Segmentation tasks. Best and second best performance are highlighted in **red** and **blue**, respectively.

| Task | Metrics | GT | LQ | BSRGAN | Real-ESRGAN | SwinIR-GAN | StableSR | DiffBIR | FaithDiff | SeeSR | SUPSR | Dream-Clear | DP2O-SR | DiT4SR | **ASASR** |
|---|---|---|---|---|---|---|---|---|---|---|---|---|---|---|---|
| *OCR* | Precision ↑ | 64.87 | 33.75 | 41.83 | 44.82 | 48.06 | 42.77 | 44.48 | 41.90 | 45.74 | 48.79 | **52.00** | 50.74 | 51.35 | **52.73** |
| | Recall ↑ | 50.32 | 3.81 | 4.11 | 7.61 | 7.31 | 21.77 | 23.74 | 31.71 | 29.33 | 36.78 | 23.95 | **40.03** | 29.98 | **45.91** |
| *Object Detection* | $AP^b$ ↑ | 48.32 | 12.53 | 15.00 | 19.36 | 18.04 | 24.66 | 23.81 | 29.52 | 28.06 | 28.04 | 27.17 | **33.51** | 28.20 | **35.62** |
| | $AP^b_{50}$ ↑ | 68.00 | 20.64 | 23.61 | 30.12 | 28.60 | 38.06 | 36.82 | 45.24 | 43.57 | 43.80 | 40.71 | **49.91** | 44.44 | **52.64** |
| | $AP^b_{75}$ ↑ | 52.91 | 12.96 | 15.74 | 20.23 | 18.86 | 25.46 | 25.20 | 30.52 | 29.34 | 29.52 | 28.69 | **36.00** | 29.18 | **38.56** |
| *Instance Segmentation* | $AP^m$ ↑ | 42.52 | 11.03 | 13.03 | 16.89 | 15.29 | 21.60 | 20.57 | 25.24 | 23.98 | 24.20 | 23.37 | **29.22** | 24.07 | **30.98** |
| | $AP^m_{50}$ ↑ | 65.29 | 18.96 | 22.08 | 28.00 | 26.32 | 35.49 | 34.08 | 41.81 | 39.56 | 40.82 | 38.34 | **46.77** | 40.34 | **49.18** |
| | $AP^m_{75}$ ↑ | 46.27 | 11.12 | 13.22 | 17.26 | 15.28 | 22.28 | 20.83 | 25.39 | 24.96 | 24.65 | 23.99 | **30.54** | 24.97 | **33.09** |
| *Semantic Segmentation* | mIoU ↑ | 49.39 | 25.18 | 21.74 | 28.85 | 25.86 | 37.31 | 34.82 | **41.72** | 39.11 | 39.16 | 36.47 | 41.54 | 37.57 | **43.33** |
| | PixAcc ↑ | 82.17 | 69.72 | 66.24 | 72.16 | 71.25 | 77.52 | 75.93 | **78.81** | 77.75 | 77.12 | 77.13 | 78.72 | 76.32 | **80.24** |

# 5. Experiments

## 5.1. Experimental Setup

**Training Datasets.** We leverage the widely adopted DIV2K (Lim et al., 2017) and LSDIR (Li et al., 2023) datasets for training. To synthesize LQ input images, we employ the higher-order degradation pipeline introduced in Real-ESRGAN (Wang et al., 2021), strictly aligning the hyperparameters with (Wu et al., 2024; Ai et al., 2024).

**Testing Datasets.** We evaluate generalization on both synthetic and real-world benchmarks. For synthetic testing, we sample 3,000 patches from DIV2K and LSDIR using the training degradation pipeline. Real-world performance is assessed on RealSR (Cai et al., 2019) and DRealSR (Wei et al., 2020). Across all protocols, HQ and LQ resolutions are standardized to $1024 \times 1024$ and $256 \times 256$, respectively.

**Metrics.** Following (Wu et al., 2024; Ai et al., 2024), we adopt PSNR and SSIM (calculated on the Y channel of trans-

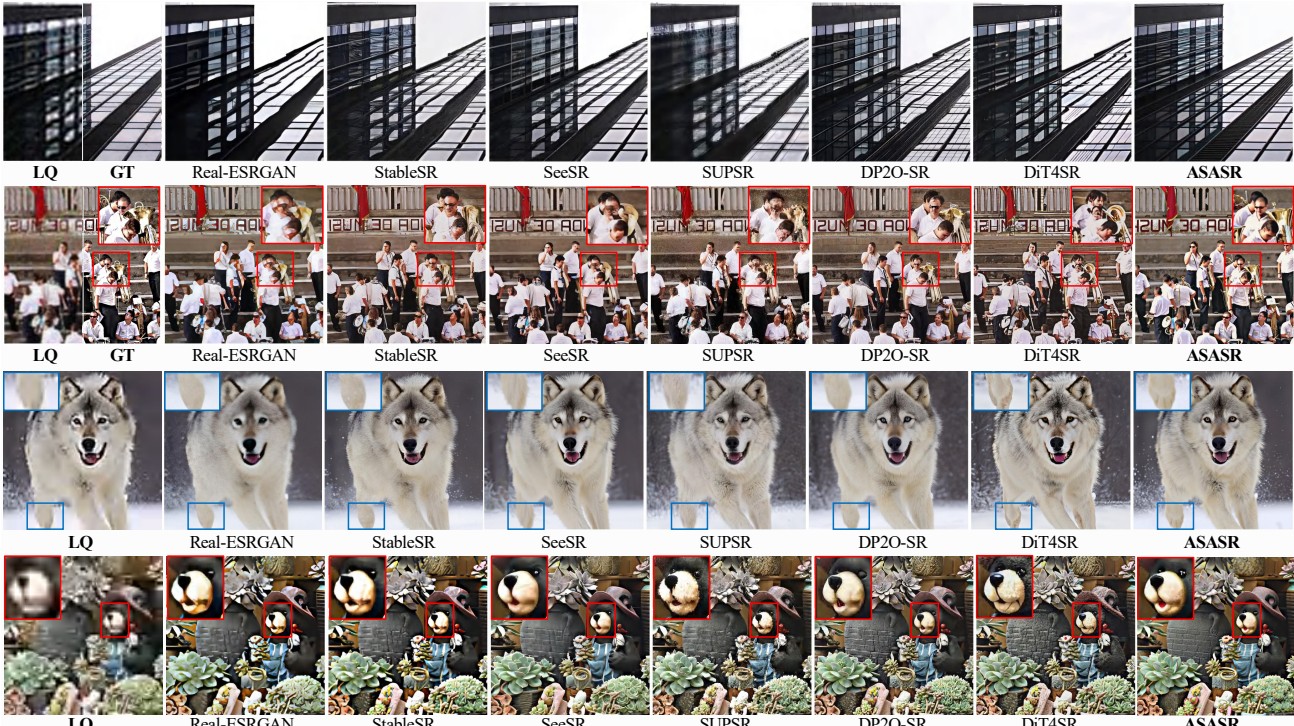

*Figure 5.* Qualitative comparisons on both synthetic (the first two rows) and real-world (the last two rows) benchmarks.

formed YCbCr space) as reference-based distortion metrics, LPIPS (Zhang et al., 2018) and DISTS (Ding et al., 2020) as reference-based perceptual metrics, MANIQA (Yang et al., 2022), MUSIQ (Ke et al., 2021) and CLIPIQA (Wang et al., 2022) as no-reference metrics.

**Baselines.** We evaluate our proposed method against state-of-the-art approaches covering diverse generative paradigms. Specifically, we compare with representative GAN-based methods, including BSRGAN (Zhang et al., 2021), Real-ESRGAN (Wang et al., 2021), and SwinIR-GAN (Liang et al., 2021). To assess performance against the most recent generative priors, we evaluate Diffusion-based models StableSR (Wang et al., 2024), DiffBIR (Lin et al., 2024), FaithDiff (Chen et al., 2024), SeeSR (Wu et al., 2024), SUPSR (Yu et al., 2024), DreamClear (Ai et al., 2024), DP2OSR (Wu et al., 2025) and DiT4SR (Duan et al., 2025).

**Implementation Details.** We perform all experiments with a scaling factor of $\times 4$. Both the ASASR and the Adversarial Network leverage the FLUX.1-dev (Labs, 2024) backbone, fine-tuned via LoRA (Hu et al., 2021) ($r = 16, \alpha = 16$). For the DPO alignment of ASASR, we adopt the AdamW optimizer with a learning rate of $1e-5$. To train the Adversarial Network, we curate a dataset by running inference with Real-ESRGAN, SeeSR, and SUPSR on a random 25% subset of DIV2K, LSDIR, RealSR, and DRealSR; this network is optimized using AdamW with a learning rate of $5e-5$. The Sobolev parameter $s$ is empirically set to 1.5.

All models are trained on 8 NVIDIA H800 GPUs. More details are detailed in Appendix C.1.

### 5.2. Comparison with State-of-the-Art Methods

**Quantitative Comparisons.** Tab. 1 presents comparisons against state-of-the-art methods. On synthetic datasets, our approach achieves superior perceptual scores (LPIPS, DISTS) and, unlike typical generative models prone to structural distortion, secures top ranks in SSIM, striking an optimal fidelity-perception balance. On real-world benchmarks, our model consistently dominates reference-free metrics (MANIQA, MUSIQ, CLIPIQA+) while maintaining leading full-reference performance. These results confirm that our method significantly outperforms competitors, producing restorations that are both photorealistic and structurally faithful to the intrinsic image manifold. Moreover, to further validate the effectiveness and generality of our method, we conduct additional experiments with different backbone models, including SD1.5 (Rombach et al., 2022) and SDXL (Podell et al., 2023). These experiments demonstrate that the performance gains of our method do not rely on the FLUX backbone. We also evaluate our method on more challenging real-world datasets, including RealLQ250 (Ai et al., 2024), which contains diverse real-world low-quality images that are absent from our training data, and Bringing Old Films Back to Life (Wan et al., 2022), a more challenging dataset consisting of degraded old film frames. The results

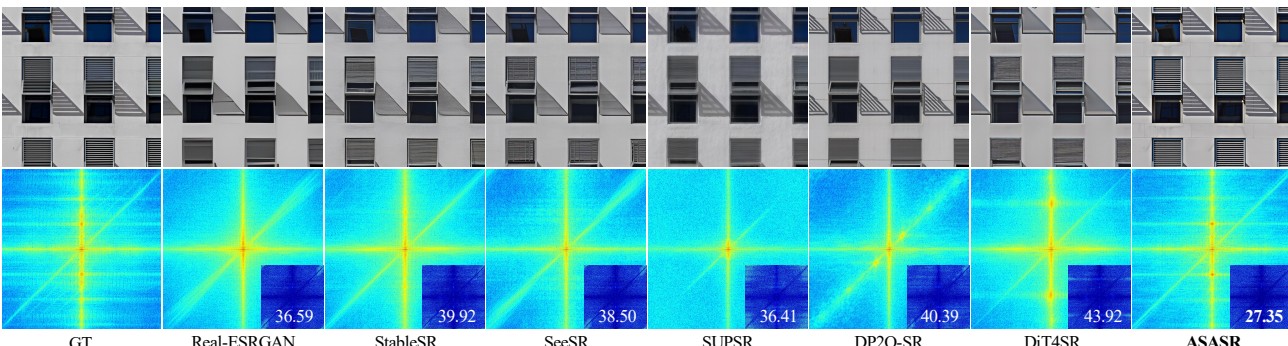

*Figure 6.* Visual and Spectral Fidelity. **Top:** Super-resolution results. **Bottom:** FFT spectra with GT-difference insets annotated with LSD scores. ASASR achieves the lowest LSD (27.35), quantitatively confirming its superior alignment with the ground truth spectral distribution, evidenced by minimal residuals compared to baselines.

*Table 3.* Ablation study on different guidance and alignment strategies. We report perceptual metrics and downstream task performance. Best values are highlighted in **bold**.

| Methods | PSNR ↑ | SSIM ↑ | LPIPS ↓ | DISTS ↓ | MANIQA ↑ | MUSIQ ↑ | CLIPIQA+ ↑ | Recall ↑ | $AP^b$ ↑ | $AP^m$ ↑ | mIoU ↑ |
|---|---|---|---|---|---|---|---|---|---|---|---|
| Sobolev Guidance | **20.60** | **0.6171** | **0.2784** | **0.2088** | **0.6519** | **71.40** | **0.7521** | **45.91** | **35.62** | **30.98** | **43.33** |
| Eulidean Guidance | 18.34 | 0.5742 | 0.3115 | 0.2337 | 0.6184 | 67.25 | 0.7196 | 41.64 | 34.18 | 29.74 | 42.15 |
| Adversarial DPO Alignment | **20.60** | **0.6171** | **0.2784** | **0.2088** | **0.6519** | **71.40** | **0.7521** | **45.91** | **35.62** | **30.98** | **43.33** |
| DPO Alignment w/ Supervised Data | 20.08 | 0.5723 | 0.3109 | 0.2345 | 0.6047 | 68.82 | 0.7163 | 40.16 | 32.49 | 28.51 | 40.26 |
| Only Supervised Learning | 19.96 | 0.5642 | 0.3135 | 0.2388 | 0.6012 | 69.15 | 0.7058 | 40.33 | 31.95 | 28.72 | 39.85 |

are provided in Appendix C.2.

**Qualitative Comparisons.** Visual comparisons in Fig. 5 demonstrate ASASR's superiority on synthetic data, where it restores sharp geometries and facial details without the aliasing seen in baselines. In real-world scenarios, it robustly handles unknown degradations, synthesizing authentic textures while suppressing artifacts. Furthermore, Fig. 6 substantiates this spectral fidelity, where ASASR achieves the lowest Log-Spectral Distance (LSD) and minimal residuals, confirming that our model faithfully reconstructs the intrinsic frequency decay of natural images. Additional comparisons are in Appendix C.4.

**User Study.** We conducted a comprehensive user study with 50 participants to evaluate restoration quality across 64 test images. Users were tasked with ranking our method against baselines based on visual naturalness and fidelity. Results shows that our model consistently outperforms all competitors achieving a Top-1 ratio of 91.1%. See Appendix C.3 for visualization and details.

**Downstream Tasks Evaluation.** To validate whether semantic information is preserved under degraded inputs, we conduct downstream evaluations on COCO (Lin et al., 2015) for object detection and instance segmentation, ADE20K (Zhou et al., 2017) for semantic segmentation, and ICDAR 2024 Occluded RoadText (George Tom et al., 2024) for OCR. We adopt representative and widely used architectures for these tasks, including Mask R-CNN (He

et al., 2018), SegFormer-B5 (Xie et al., 2021), and PaddleOCR v3 (Li et al., 2022), respectively. For evaluation, COCO images are standardized to $512 \times 512$, while cropping is applied to ADE20K and a 2,000-frame subset of ICDAR to better preserve local structures and fine-grained textual details. Using the identical degradation pipeline as in training, our method achieves SOTA performance across all metrics (Tab. 2), indicating stronger fidelity in preserving high-level semantic structures under challenging degradations. Detailed quantitative results and additional analyses are provided in Appendix C.4.

### 5.3. Ablation Study

We perform ablation studies to scrutinize the contributions of our guidance and alignment mechanisms, as detailed in Tab. 3. Across perceptual (PSNR, SSIM, LPIPS, DISTS), quality (MANIQA, MUSIQ, CLIPIQA+), and semantic metrics (Recall, AP, mIoU), our full method consistently outperforms ablated versions. Regarding guidance, *Sobolev Guidance* (Full Model) yields significantly superior results compared to *Euclidean Guidance* (w/o SSR), validating its optimization effectiveness. Similarly, for alignment, *Adversarial DPO* (Full Model) demonstrates a clear advantage over both *Only Supervised Learning* (w/o SSR&AMG) and standard *DPO w/ Supervised Data* (w/o AMG). The comparison among these variants highlights the coupling between SSR and AMG: AMG supplies realism-oriented alignment, whereas SSR provides a spectrally aware ge-

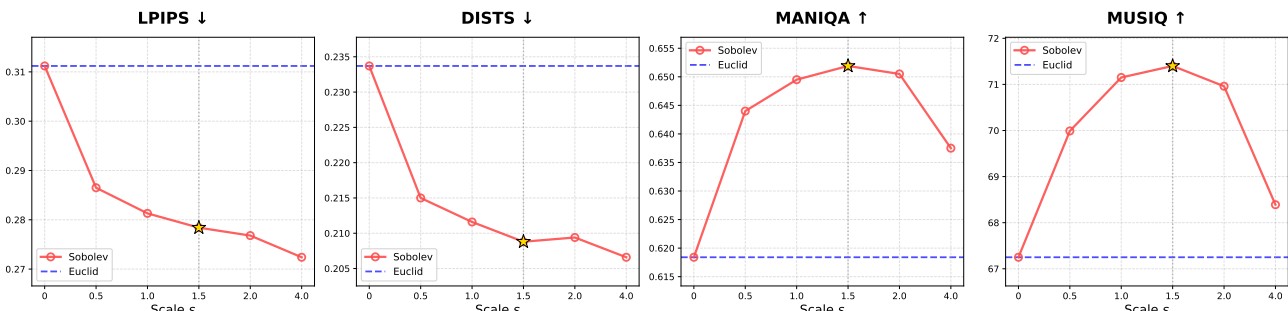

*Figure 7.* Impact analysis on the Sobolev index $s$. We select $s = 1.5$ (marked by star) as the optimal trade-off between structural fidelity and texture realism.

ometry that regularizes its optimization trajectory. As a result, applying AMG without SSR may introduce aggressive high-frequency details that benefit perceptual realism less consistently and can impair distortion-oriented metrics, while their combination enables alignment to improve perceptual quality without sacrificing reconstruction fidelity. Notably, substantial gains in downstream tasks confirm that our strategies enhance photorealism while preserving the semantic integrity essential for high-level vision.

In addition, we conduct sensitivity analysis on the Sobolev index $s$. As shown in Fig. 7, while stronger regularity ($s \geq 2$) benefits reference-based metrics via aggressive smoothing, it degrades perceptual quality by erasing high-frequency textures. We thus adopt $s = 1.5$ as the optimal equilibrium between fidelity and realism.

## 6. Related Work

The evolution of image super-resolution has shifted from GAN-based training (Zhang et al., 2021; Wang et al., 2021; Liang et al., 2021) toward exploiting large-scale generative priors (Wang et al., 2024; Lin et al., 2024; Wu et al., 2024; Yu et al., 2024), which provide stronger natural image priors for blind and real-world restoration. This paradigm now encompasses diverse architectures, including Diffusion Transformers (Cheng et al., 2024; Duan et al., 2025), autoregressive models (Qu et al., 2025), as well as Gaussian (Hu et al., 2024) and spiking (Xiao et al., 2025) formulations, reflecting a broad trend toward leveraging more expressive generative backbones. Recent advances further improve practical usability through efficient distillation (Xu et al., 2025; Li et al., 2025a), while semantic or textual guidance has been introduced to enhance controllability and preserve high-level content (Chen et al., 2025; Hu et al., 2025; Zuo et al., 2025). Despite their impressive perceptual quality, generative SR models remain prone to hallucinated textures and structurally inconsistent details, especially under severe or out-of-distribution degradations. To address this issue, recent studies explore alignment strategies (Chen et al., 2024;

Li et al., 2025b) and DPO-style adaptations that encourage outputs to better match human or perceptual preferences. Notably, DP2O-SR (Wu et al., 2025) steers generation via aggregated IQA metrics; however, such heuristic optimization lacks a principled theoretical foundation, disconnecting the proxy objective from the intrinsic geometric structure of the natural image manifold.

## 7. Conclusion

We propose ASASR to bridge the spectral gap between isotropic generative priors and the natural image manifold. By enforcing Sobolev Spectral Rectification, our method constrains optimization within a Riemannian geometry that respects the characteristic spectral decay of natural images, thereby promoting perceptually plausible yet structurally faithful restoration. This process is driven by a Riesz-grounded adversary, which leverages worst-case Sobolev gradients to identify and rectify structural deviations that are often overlooked by conventional pixel- or perception-level objectives. Extensive experiments demonstrate that ASASR achieves a superior fidelity-realism balance across diverse degradations and backbones. More broadly, our framework provides a rigorous spectral-geometric blueprint for addressing hallucination and alignment challenges in ill-posed inverse problems.

## Impact Statement

This work presents a robust framework for high-fidelity image super-resolution with significant potential for consumer imaging deployment. By effectively mitigating spectral artifacts, our method empowers mobile imaging systems to overcome hardware limitations like digital zoom and low-light noise. Furthermore, as user-generated content platforms (e.g., photo-logs) expand, our approach serves as a vital cloud-based tool to revitalize compressed uploads. By democratizing access to professional-grade restoration, we aim to preserve the visual integrity of digital archives and foster a high-quality digital ecosystem.

## Acknowledgement

This work was supported by National Natural Science Foundation of China (Grant Nos. 62425606, 62576342, 62550062, 32341009), Beijing Natural Science Foundation (4252054, L257008), Beijing Nova Program (20230484276, 20240484601). These contributions have been instrumental in enabling the advancement of this work.

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

# A. Derivation of the S-DPO Objective

In this section, we rigorously derive the S-DPO objective. We first establish the velocity-space probabilistic formulation to recover the standard $\ell_2$ preference objective, ensuring stability in the continuous-time limit. We then introduce the Sobolev Spectral Rectification to generalize this into the frequency-aware S-DPO loss, while addressing the theoretical considerations regarding boundary conditions.

## A.1. Continuous-Time Velocity-Space Formulation

To adapt Direct Preference Optimization (DPO) to the continuous Flow Matching framework without succumbing to the vanishing Signal-to-Noise Ratio (SNR) inherent in discrete SDE approximations, we model the generative process as a probabilistic policy defined on the tangent bundle (velocity space).

Standard Flow Matching defines the generative process via an ODE: $\mathrm{d}\boldsymbol{x}_t = \boldsymbol{v}(\boldsymbol{x}_t, t)\mathrm{d}t$, targeting the conditional vector field $\boldsymbol{u}_t(\boldsymbol{x}|\boldsymbol{x}_1) = (\boldsymbol{x}_1 - \boldsymbol{x}_t)/(1-t)$. We define the policy $p_\theta(\cdot|\boldsymbol{x}_t)$ as an isotropic Gaussian distribution over the instantaneous velocity $\boldsymbol{v} \in \mathbb{R}^d$, centered at the model prediction $\boldsymbol{v}_\theta$:

$$p_\theta(\boldsymbol{v}|\boldsymbol{x}_t, \boldsymbol{c}) := \mathcal{N}\left(\boldsymbol{v}; \boldsymbol{v}_\theta(\boldsymbol{x}_t, t, \boldsymbol{c}), \eta^2\mathbf{I}\right), \tag{21}$$

where $\eta^2$ is a temperature parameter. In this framework, the "optimal action" is the deterministic target velocity $\boldsymbol{u}_t$.

The instantaneous log-likelihood ratio at time $t$ compares the probability of the optimal velocity $\boldsymbol{u}_t$ under the policy $p_\theta$ versus the reference $p_{\text{ref}}$:

$$\mathcal{R}_t(\boldsymbol{x}_t) := \log \frac{p_\theta(\boldsymbol{u}_t|\boldsymbol{x}_t, \boldsymbol{c})}{p_{\text{ref}}(\boldsymbol{u}_t|\boldsymbol{x}_t, \boldsymbol{c})}, \tag{22}$$

Expanding the Gaussian densities, the normalization constants cancel out, leaving the difference of squared Euclidean norms:

$$\begin{aligned}
\mathcal{R}_t(\boldsymbol{x}_t) &= \left(-\frac{1}{2\eta^2}\|\boldsymbol{u}_t - \boldsymbol{v}_\theta\|_2^2\right) - \left(-\frac{1}{2\eta^2}\|\boldsymbol{u}_t - \boldsymbol{v}_{\text{ref}}\|_2^2\right) \\
&= \frac{1}{2\eta^2}\left(\|\boldsymbol{v}_{\text{ref}} - \boldsymbol{u}_t\|_2^2 - \|\boldsymbol{v}_\theta - \boldsymbol{u}_t\|_2^2\right),
\end{aligned} \tag{23}$$

Let $\boldsymbol{\gamma}_\psi := \boldsymbol{v}_\psi(\boldsymbol{x}_t) - \boldsymbol{u}_t$ denote the vector field residual. Since the target $\boldsymbol{u}_t$ is deterministic (derived from boundary conditions), this evaluation is free from stochastic transition noise, ensuring numerical stability as $\Delta t \to 0$. Absorbing constants into the proportionality, we recover the $\ell_2$-based objective:

$$\log \frac{p_\theta(\boldsymbol{x}_{t-\Delta t}|\boldsymbol{x}_t, \boldsymbol{c})}{p_{\text{ref}}(\boldsymbol{x}_{t-\Delta t}|\boldsymbol{x}_t, \boldsymbol{c})} \propto \mathcal{R}_t(\boldsymbol{x}_t) \propto -\left(\|\boldsymbol{\gamma}_\theta\|_2^2 - \|\boldsymbol{\gamma}_{\text{ref}}\|_2^2\right), \tag{24}$$

## A.2. Sobolev Spectral Rectification and S-DPO

To mitigate the frequency-agnostic nature of the $\ell_2$ norm, we propose *Sobolev Spectral Rectification*. We generalize the isotropic assumption by replacing the scalar variance $\eta^2\mathbf{I}$ with a structured spectral covariance $\boldsymbol{\Sigma}_s$.

In our framework, the reference policy is parameterized on the Sobolev manifold. We define the likelihood of the reference policy $p_{\text{ref}}$ directly in this space:

$$p_{\text{ref}}(\boldsymbol{v}|\boldsymbol{x}_t) := \mathcal{N}(\boldsymbol{v}; \boldsymbol{v}_{\text{ref}}(\boldsymbol{x}_t), \eta^2\boldsymbol{\Sigma}_s), \tag{25}$$

This formulation ensures that the trained policy $p_\theta$ and the reference $p_{\text{ref}}$ share the same support and curvature defined by $\boldsymbol{\Sigma}_s$. Consequently, the regularization in S-DPO penalizes deviations from the reference weights weighted by their spectral importance, rather than Euclidean distance.

The policy is now modeled as $p_\theta(\boldsymbol{v}) = \mathcal{N}(\boldsymbol{v}; \boldsymbol{v}_\theta, \eta^2\boldsymbol{\Sigma}_s)$. The determinant term $\det(2\pi\eta^2\boldsymbol{\Sigma}_s)$ depends only on fixed hyperparameters and cancels out in the ratio. The log-likelihood ratio becomes the difference of Mahalanobis distances:

$$\mathcal{R}_t^{H^s}(\boldsymbol{x}_t) = -\frac{1}{2\eta^2}\left(\|\boldsymbol{\gamma}_\theta\|_{\boldsymbol{\Sigma}_s^{-1}}^2 - \|\boldsymbol{\gamma}_{\text{ref}}\|_{\boldsymbol{\Sigma}_s^{-1}}^2\right), \tag{26}$$

Recall that $\boldsymbol{\Sigma}_s = \mathcal{F}^{-1}\mathbf{D}_s\mathcal{F}$, where $\mathbf{D}_s(\boldsymbol{\omega}) = (1 + \|\boldsymbol{\omega}\|_2^2)^{-s}$. The inverse operator is $\boldsymbol{\Sigma}_s^{-1} = \mathcal{F}^{-1}\mathbf{D}_s^{-1}\mathcal{F}$. By Plancherel's theorem, the quadratic form for any residual $\boldsymbol{\gamma}$ is:

$$\|\boldsymbol{\gamma}\|_{\boldsymbol{\Sigma}_s^{-1}}^2 = \boldsymbol{\gamma}^\top \mathcal{F}^{-1}\mathbf{D}_s^{-1}\mathcal{F}\boldsymbol{\gamma} = (\mathcal{F}\boldsymbol{\gamma})^{\mathcal{H}}\mathbf{D}_s^{-1}(\mathcal{F}\boldsymbol{\gamma})$$
$$= \sum_{\boldsymbol{\omega}} (1 + \|\boldsymbol{\omega}\|_2^2)^s \cdot |\hat{\boldsymbol{\gamma}}(\boldsymbol{\omega})|^2 \equiv \|\boldsymbol{\gamma}\|_{H^s}^2, \tag{27}$$

Substituting the Sobolev norm equivalence back into the ratio, we define the Sobolev Energy Gap:

$$\Delta\mathcal{E}_{H^s}(\boldsymbol{x}_t, \boldsymbol{c}) := \|\boldsymbol{\gamma}_\theta\|_{H^s}^2 - \|\boldsymbol{\gamma}_{\text{ref}}\|_{H^s}^2, \tag{28}$$

Finally, incorporating the standard DPO logistic loss structure over the preference dataset $\mathcal{D}$ (where winners $\boldsymbol{x}^w$ should have higher likelihood than losers $\boldsymbol{x}^l$), we arrive at the S-DPO objective:

$$\mathcal{L}_{\text{S-DPO}}(\theta) = -\mathbb{E}_{(\boldsymbol{c}, \boldsymbol{x}_1^w, \boldsymbol{x}_1^l) \sim \mathcal{D}, t \sim \mathcal{U}(0,T)}\Big[\log \sigma\Big(\beta\Big[\Delta\mathcal{E}_{H^s}(\boldsymbol{x}_t^l, \boldsymbol{c}) - \Delta\mathcal{E}_{H^s}(\boldsymbol{x}_t^w, \boldsymbol{c})\Big]\Big)\Big], \tag{29}$$

Note that the sign inversion inside the sigmoid ($\boldsymbol{x}^l - \boldsymbol{x}^w$) arises because we minimize the energy (residual) to maximize likelihood.

## B. Proof of Proposition in Adversarial Manifold Guidance

### B.1. Proof of Proposition 1: Sobolev Geometry of Hard Negatives

In this section, we provide the rigorous derivation for the optimal hard negative perturbation presented in Proposition 1. As argued in Section 4.2, the adversary targets the model's *spectral blind spots* by seeking perturbations that minimize the Euclidean residual energy (mimicking the model's intrinsic bias) while strictly adhering to a structural trust region defined by the Sobolev norm.

We seek a perturbation $\boldsymbol{\delta}_t$ that induces structural deviation yet remains indistinguishable to the isotropic baseline. Mathematically, this corresponds to finding a state within a Sobolev trust region $\|\boldsymbol{\delta}_t\|_{H^s} \leq \varepsilon_t$ that minimizes the model's Euclidean residual energy $\mathcal{J}_{L^2}$. This optimization targets the model's null space, identifying artifacts that improperly minimize the training objective. The problem is formulated as:

$$\min_{\boldsymbol{\delta}_t} \quad \mathcal{J}_{L^2}(\boldsymbol{x}_t^w + \boldsymbol{\delta}_t) \quad \text{s.t.} \quad \|\boldsymbol{\delta}_t\|_{H^s} \leq \varepsilon_t, \tag{30}$$

Consider the first-order Taylor expansion of the objective around $\boldsymbol{x}_t^w$:

$$\mathcal{J}_{L^2}(\boldsymbol{x}_t^w + \boldsymbol{\delta}_t) \approx \mathcal{J}_{L^2}(\boldsymbol{x}_t^w) + \langle \nabla_{\boldsymbol{x}}\mathcal{J}_{L^2}, \boldsymbol{\delta}_t \rangle_{L^2}, \tag{31}$$

where $\nabla_{\boldsymbol{x}}\mathcal{J}_{L^2}$ is the standard Euclidean gradient. Since the zero-order term is constant, the problem reduces to minimizing the linear alignment term $\langle \nabla_{\boldsymbol{x}}\mathcal{J}_{L^2}, \boldsymbol{\delta}_t \rangle_{L^2}$ subject to the constraint.

Crucially, the constraint is governed by the Sobolev inner product. Using the spectral operator definition $\boldsymbol{\Sigma}_s = \mathcal{F}^{-1}\mathbf{D}_s\mathcal{F}$, the constraint can be rewritten in terms of the $L^2$ inner product via the inverse operator $\boldsymbol{\Sigma}_s^{-1}$:

$$\|\boldsymbol{\delta}_t\|_{H^s}^2 = \langle \boldsymbol{\delta}_t, \boldsymbol{\delta}_t \rangle_{H^s} = \langle \boldsymbol{\delta}_t, \boldsymbol{\Sigma}_s^{-1}\boldsymbol{\delta}_t \rangle_{L^2} \leq \varepsilon_t^2, \tag{32}$$

To solve this constrained optimization, we construct the Lagrangian $\mathcal{L}(\boldsymbol{\delta}_t, \lambda)$ with a Lagrange multiplier $\lambda > 0$:

$$\mathcal{L}(\boldsymbol{\delta}_t, \lambda) = \langle \nabla_{\boldsymbol{x}}\mathcal{J}_{L^2}, \boldsymbol{\delta}_t \rangle_{L^2} + \frac{\lambda}{2}\left(\langle \boldsymbol{\delta}_t, \boldsymbol{\Sigma}_s^{-1}\boldsymbol{\delta}_t \rangle_{L^2} - \varepsilon_t^2\right), \tag{33}$$

Taking the Fréchet derivative with respect to $\boldsymbol{\delta}_t$ and setting it to zero:

$$\frac{\partial\mathcal{L}}{\partial\boldsymbol{\delta}_t} = \nabla_{\boldsymbol{x}}\mathcal{J}_{L^2} + \lambda\boldsymbol{\Sigma}_s^{-1}\boldsymbol{\delta}_t^* = 0, \tag{34}$$

Since $\boldsymbol{\Sigma}_s^{-1}$ is a symmetric positive-definite operator, it is invertible. We solve for the optimal unnormalized direction:

$$\boldsymbol{\delta}_t^* = -\frac{1}{\lambda}\boldsymbol{\Sigma}_s \nabla_{\boldsymbol{x}}\mathcal{J}_{L^2}, \tag{35}$$

To determine the scalar multiplier $\lambda$, we invoke the active constraint condition $\|\boldsymbol{\delta}_t^*\|_{H^s} = \varepsilon_t$. Substituting Eq. (35) into the squared norm:

$$\begin{aligned}
\|\boldsymbol{\delta}_t^*\|_{H^s}^2 &= \langle \boldsymbol{\delta}_t^*, \boldsymbol{\Sigma}_s^{-1}\boldsymbol{\delta}_t^* \rangle_{L^2} \\
&= \frac{1}{\lambda^2}\langle \boldsymbol{\Sigma}_s \nabla_{\boldsymbol{x}}\mathcal{J}_{L^2}, \boldsymbol{\Sigma}_s^{-1}\boldsymbol{\Sigma}_s \nabla_{\boldsymbol{x}}\mathcal{J}_{L^2} \rangle_{L^2} \\
&= \frac{1}{\lambda^2}\langle \nabla_{\boldsymbol{x}}\mathcal{J}_{L^2}, \boldsymbol{\Sigma}_s \nabla_{\boldsymbol{x}}\mathcal{J}_{L^2} \rangle_{L^2},
\end{aligned} \tag{36}$$

Solving for $1/\lambda$ yields the scaling factor $\varepsilon_t/\|\boldsymbol{\Sigma}_s \nabla_{\boldsymbol{x}}\mathcal{J}_{L^2}\|_{H^s}$. Substituting this back gives the closed-form solution:

$$\boldsymbol{\delta}_t^* = -\varepsilon_t \frac{\boldsymbol{\Sigma}_s \nabla_{\boldsymbol{x}}\mathcal{J}_{L^2}}{\sqrt{\langle \nabla_{\boldsymbol{x}}\mathcal{J}_{L^2}, \boldsymbol{\Sigma}_s \nabla_{\boldsymbol{x}}\mathcal{J}_{L^2} \rangle_{L^2}}}, \tag{37}$$

**Geometric Interpretation and the Role of Preconditioning.** This result formally elucidates the role of $\boldsymbol{\Sigma}_s$ as a spectral preconditioner. The term $\boldsymbol{\Sigma}_s \nabla_{\boldsymbol{x}}\mathcal{J}_{L^2}$ corresponds to the *Natural Gradient* on the Sobolev manifold. Mathematically, the Euclidean gradient $\nabla_{\boldsymbol{x}}\mathcal{J}_{L^2}$ resides in the dual space and often contains high-frequency stochastic noise. By the Riesz Representation Theorem, $\boldsymbol{\Sigma}_s$ acts as the canonical isomorphism mapping this irregular dual vector back to the primal space $H^s$.

Crucially, this preconditioning does not merely "smooth" the image in a naive sense; rather, it enforces *spectral regularity*. Even if the raw gradient points towards unstructured noise (a common property of neural network gradients), the operator $\boldsymbol{\Sigma}_s$ filters these incoherent components. This ensures that the resulting perturbation $\boldsymbol{\delta}_t^*$ manifests as **structurally consistent degradations**—such as recurring artifacts or texture distortions—rather than imperceptible noise. This justifies the use of our parametric adversary $\mathcal{A}_\phi$ (Proposition 2), which learns to approximate these coherent failure patterns to effectively penalize model hallucinations.

### B.2. Proof of Proposition 2: Theoretical Consistency of the Parametric Adversary

In this section, we provide the rigorous proof for Proposition 2, demonstrating that a parametric adversary $\mathcal{A}_\phi$ trained to minimize the expected residual energy under a Sobolev constraint implicitly recovers the optimal Sobolev descent direction derived in Proposition 1.

**Optimization in Velocity Space.** In our Flow Matching formulation, the adversary $\mathcal{A}_\phi$ is parameterized as a correction to the *velocity field* $\boldsymbol{v}_\theta$. While Proposition 1 characterizes the optimal descent direction $\boldsymbol{\delta}_t$ in the state space, the parametric realization requires translating this geometric guidance into the velocity domain. To ensure numerical stability across the entire time horizon $t \in [0,1]$ and avoid singularities associated with time-dependent Jacobians near the boundaries, we impose the Sobolev trust region directly on the velocity potential. Consequently, the optimization problem is formalized as:

$$\min_\phi \quad \mathbb{E}_{\boldsymbol{x}_t}\left[\mathcal{J}(\boldsymbol{x}_t + \Delta\boldsymbol{x}(\mathcal{A}_\phi))\right] \quad \text{s.t.} \quad \|\mathcal{A}_\phi(\boldsymbol{x}_t)\|_{H^s} \leq \varepsilon, \tag{38}$$

where $\Delta\boldsymbol{x}(\mathcal{A}_\phi)$ denotes the state deviation induced by the velocity perturbation $\mathcal{A}_\phi$ over a discretization step $\Delta t$.

**Linearization via Euler Integration.** Consider the first-order Taylor expansion of the energy function for a small step $\Delta t$:

$$\mathcal{J}(\boldsymbol{x}_t + \Delta\boldsymbol{x}) \approx \mathcal{J}(\boldsymbol{x}_t) + \langle \nabla_{\boldsymbol{x}}\mathcal{J}, \Delta\boldsymbol{x} \rangle_{L^2}, \tag{39}$$

Under the standard Euler integration scheme, the velocity perturbation $\mathcal{A}_\phi$ induces a linear state update $\Delta\boldsymbol{x} \approx \mathcal{A}_\phi \cdot \Delta t$. Substituting this relationship into the expansion, and noting that $\mathcal{J}(\boldsymbol{x}_t)$ is independent of $\phi$, the objective reduces to minimizing the inner product:

$$\min_\phi \quad \mathbb{E}_{\boldsymbol{x}_t}\left[\langle \nabla_{\boldsymbol{x}}\mathcal{J}, \mathcal{A}_\phi \cdot \Delta t \rangle_{L^2}\right] \quad \text{s.t.} \quad \|\mathcal{A}_\phi\|_{H^s} \leq \varepsilon, \tag{40}$$

Assuming sufficient representational capacity (i.e., $\mathcal{A}_\phi$ acts as a universal approximator), minimizing the expected loss is equivalent to minimizing the objective pointwise for each $x_t$. Absorbing the scalar $\Delta t$ into the effective step size, we solve the following constrained variational problem for a fixed state:

$$\mathcal{A}_{\phi^*} = \arg\min_{v \in H^s} \langle \nabla_x \mathcal{J}, v \rangle_{L^2} \quad \text{s.t.} \quad \|v\|_{H^s} \leq \varepsilon, \tag{41}$$

**Solution via Spectral Duality.** To solve this, we leverage the duality between the $L^2$ and $H^s$ spaces. Using the definition of the Sobolev inner product $\langle u, v \rangle_{L^2} = \langle \Sigma_s u, v \rangle_{H^s}$, we rewrite the objective functional:

$$\langle \nabla_x \mathcal{J}, v \rangle_{L^2} = \langle \Sigma_s \nabla_x \mathcal{J}, v \rangle_{H^s}, \tag{42}$$

The term $\nabla_x^{H^s} \mathcal{J} := \Sigma_s \nabla_x \mathcal{J}$ can be interpreted as the *natural Sobolev gradient*. The optimization is now formulated entirely within the geometry of the Hilbert space $H^s$:

$$\min_v \langle \nabla_x^{H^s} \mathcal{J}, v \rangle_{H^s} \quad \text{s.t.} \quad \|v\|_{H^s} \leq \varepsilon, \tag{43}$$

By the Cauchy-Schwarz inequality in Hilbert spaces, the inner product is minimized when $v$ is strictly *anti-parallel* to the gradient direction and lies on the boundary of the feasible set. Therefore, the optimal solution is:

$$\mathcal{A}_{\phi^*} = -\varepsilon \frac{\Sigma_s \nabla_x \mathcal{J}}{\|\Sigma_s \nabla_x \mathcal{J}\|_{H^s}}, \tag{44}$$

**Equivalence to Proposition 1.** Finally, we verify that this velocity-based solution matches the state perturbation derived in Proposition 1. We expand the Sobolev norm in the denominator:

$$
\begin{aligned}
\|\Sigma_s \nabla_x \mathcal{J}\|_{H^s}^2 &= \langle \Sigma_s \nabla_x \mathcal{J}, \Sigma_s \nabla_x \mathcal{J} \rangle_{H^s} \\
&= \langle \Sigma_s \nabla_x \mathcal{J}, \Sigma_s^{-1}(\Sigma_s \nabla_x \mathcal{J}) \rangle_{L^2} \quad (\text{via } \Sigma_s^{-1}) \\
&= \langle \Sigma_s \nabla_x \mathcal{J}, \nabla_x \mathcal{J} \rangle_{L^2} \\
&= \langle \nabla_x \mathcal{J}, \Sigma_s \nabla_x \mathcal{J} \rangle_{L^2} \quad (\text{symmetry of } \Sigma_s),
\end{aligned}
\tag{45}
$$

Substituting this norm back into Eq. (44) yields:

$$\mathcal{A}_{\phi^*} = -\varepsilon \frac{\Sigma_s \nabla_x \mathcal{J}}{\sqrt{\langle \nabla_x \mathcal{J}, \Sigma_s \nabla_x \mathcal{J} \rangle_{L^2}}}, \tag{46}$$

This expression is identical to Eq. (19) in Proposition 1. This confirms that minimizing the energy loss with a bounded velocity network implicitly learns the optimal Sobolev-preconditioned descent direction.

### B.3. Empirical Examination of the Capacity Assumption

Prop. 4.2 characterizes the Sobolev-optimal perturbation in function space, while the sufficient-capacity assumption is only used to connect this functional optimum to its neural parameterization. To examine whether this assumption is reasonable in practice, we vary the capacity of the adversary by changing the LoRA size and report the resulting performance in Tab. 4. The results show that increasing adversary capacity from very small adapters yields clear gains, while performance largely saturates once the adapter becomes moderately expressive. This behavior is consistent with Prop. 4.2: the adversary does not require arbitrarily large capacity, but only sufficient expressiveness to realize or closely approximate the Sobolev-optimal direction.

## C. More Experiments

### C.1. More Implementation Details

Supplementing the experimental configurations outlined in the main text, we provide further engineering specifics to facilitate reproducibility. To optimize training throughput and memory efficiency, we employ *BF16* mixed-precision training

*Table 4.* Effect of adversary capacity under different LoRA sizes. Performance improves with capacity in the small-adapter regime and largely saturates beyond moderate sizes.

| Adversary Capacity | Params. | PSNR ↑ | SSIM ↑ | MANIQA ↑ | CLIPIQA+ ↑ |
|---|---|---|---|---|---|
| $(4, 4)$ | ∼7.4M | 19.43 | 0.6026 | 0.6328 | 0.7349 |
| $(8, 8)$ | ∼14.8M | 20.54 | 0.6158 | 0.6487 | 0.7498 |
| $(16, 16)$ | ∼29.5M | 20.60 | **0.6171** | **0.6519** | 0.7521 |
| $(32, 32)$ | ∼59.0M | **20.61** | 0.6170 | 0.6517 | 0.7524 |
| $(64, 64)$ | ∼118.0M | 20.58 | 0.6167 | 0.6515 | **0.7525** |

with gradient checkpointing enabled on the FLUX.1-dev backbone. For the inference stage, we adopt a 28-step sampling schedule to balance generation quality and latency.

A component of our training infrastructure is the management of the reference policy required by the DPO objective. To instantiate the reference model $p_{\text{ref}}$ without duplicating the massive parameters of FLUX.1-dev, we adopt a Dual-LoRA strategy. Specifically, we maintain the pre-trained SFT weights as a frozen LoRA adapter ($\mathcal{M}_{\text{SFT}}$) to define the reference distribution. Simultaneously, we introduce a separate, zero-initialized LoRA adapter ($\mathcal{M}_{\text{DPO}}$) to parameterize the policy update. Thus, the learning process starts explicitly from the SFT baseline, ensuring optimization stability. The final inference weights are the algebraic sum of the base parameters and both adapters.

To better accommodate the non-periodic nature of natural images, we implement the spectral operator $\mathcal{F}$ using the Discrete Cosine Transform (DCT-II). By implicitly enforcing Neumann boundary conditions via symmetric reflection, the DCT basis avoids potential boundary discontinuities associated with the periodic assumption of the DFT. This ensures better continuity at the image boundaries and maintains the regularity consistent with the Sobolev norm $\| \cdot \|_{H^s}$.

Optimization is conducted with an effective global batch size of 512, achieved via a per-device batch size of 8 and 8 gradient accumulation steps across the GPU cluster. We utilize a linear warmup of 200 steps to stabilize early training dynamics, and the DPO KL penalty coefficient $\beta$ is set to 2000. Finally, regarding data processing, all input images are normalized to $[-1, 1]$, and we use empty text prompts during training to force the model to rely exclusively on image-conditional cues for restoration.

*Table 5.* Performance Analysis on NVIDIA A800. We evaluate the memory efficiency, latency, and Model Flops Utilization (MFU) of the FLUX.1 DEV model during the super-resolution task. The metrics are averaged over 100 inference runs excluding warm-up.

| Metric | Value | Unit | Description |
|---|---|---|---|
| Avg. Latency | 19.60 | s/img | Inference time per image |
| Peak VRAM | 33.02 | GB | Max memory usage (96.2% Eff.) |
| Total Compute | 2.85 | PFLOPs | Total floating-point ops per img |
| MFU | 46.6 | % | Model Flops Utilization (Est.) |

Furthermore, we also conduct performance experiments. Tab. 5 presents the quantitative analysis of the inference efficiency. Our implementation achieves a high Model Flops Utilization (MFU) of 46.6%, indicating effective hardware saturation on the A800 GPU. The peak VRAM usage reaches 33.02 GB with a 96.2% memory efficiency, demonstrating that our system maximizes the available memory bandwidth. With an average latency of 19.60 seconds per image, the system maintains a practical throughput for high-resolution generation tasks.

### C.2. More Quantitative Experiments

To further examine the generality of our method across different generative backbones, we conduct additional experiments using SD1.5 and SDXL, as shown in Tab. 6. The consistent improvements over the corresponding SFT baselines indicate that the effectiveness of our method is not specific to the FLUX backbone.

In addition, to evaluate the out-of-distribution generalization ability of our method, we further test on two more challenging real-world datasets: RealLQ250 and Bringing Old Films Back to Life. As shown in Tab. 7 and Tab. 8, our method achieves competitive or superior performance across multiple non-reference image quality metrics, demonstrating its robustness on diverse real-world degradations beyond the training distribution.

*Table 6.* Quantitative comparison of different backbones and methods.

| Backbone | Methods | PSNR↑ | SSIM↑ | MANIQA↑ | CLIPIQA+↑ |
|---|---|---|---|---|---|
| SD1.5 | SFT | 18.52 | 0.4873 | 0.4236 | 0.6142 |
| SD1.5 | +SSR+AMG | **19.14** | **0.5207** | **0.4891** | **0.6658** |
| SDXL | SFT | 19.31 | 0.5284 | 0.5473 | 0.6824 |
| SDXL | +SSR+AMG | **19.92** | **0.5681** | **0.6104** | **0.7289** |
| FLUX | SFT | 19.96 | 0.5642 | 0.6012 | 0.7058 |
| **FLUX (ours)** | +SSR+AMG | **20.60** | **0.6171** | **0.6519** | **0.7521** |

*Table 7.* Quantitative comparison on RealLQ250. We compare ASASR with representative real-world image super-resolution methods using non-reference image quality assessment metrics.

| Methods | NIQE↓ | MANIQA↑ | MUSIQ↑ | CLIPIQA+↑ |
|---|---|---|---|---|
| DP2O-SR | 4.02 | 0.4494 | 69.87 | 0.7042 |
| SeeSR | 4.41 | 0.4992 | 70.57 | 0.7104 |
| DreamClear | **3.55** | 0.4351 | 66.76 | 0.7116 |
| DiT4SR | 4.35 | 0.4094 | 63.45 | 0.6829 |
| **ASASR** | 3.64 | **0.5168** | **71.16** | **0.7202** |

## C.3. User Study

To conduct a holistic evaluation of restoration quality, we organized a user study involving 50 participants to assess perceptual realism and semantic fidelity. We randomly sampled 64 low-quality images from the test sets, processing them with ASASR and six competitive baselines (DiT4SR, DP2OSR, SUPIR, SeeSR, StableSR, RealESRGAN) to generate comparative groups. For each of the 64 test images, participants were presented with the low-quality reference and asked to rank the restored images within each group based on visual naturalness, detail precision, and the absence of artifacts. To ensure fairness, the method labels were anonymized, and the initial display order was randomized.

To quantitatively analyze the subjective feedback, we adopted two metrics: Vote Percentage (Top-1 Ratio) and Top-K Ratio. The former measures the frequency with which a method is ranked first. For the latter, we calculate the frequency that a method $i$ appears within the top-$k$ rankings. The Top-K ratio $R_i^k$ is formally defined as:

$$R_i^k = \frac{1}{N} \sum_{j=1}^{N} \mathbb{K}(\text{rank}_{ij} \leq k), \tag{47}$$

where $N$ denotes the total number of evaluation groups, $\text{rank}_{ij}$ represents the ranking of method $i$ in the $j$-th group (where 1 indicates the best), and $\mathbb{K}(\cdot)$ is the indicator function.

As illustrated in Fig. 8, ASASR outperforms all baselines by a substantial margin. In the Top-1 Ratio analysis (Fig. 8(a)), our method secured 91.1% of the first-place votes, indicating an overwhelming user preference. Furthermore, the Top-K Ratio results (Fig. 8(b)) demonstrate that ASASR is consistently ranked among the top choices across all $k$ levels, highlighting the stability and reliability of our geometry-aware restoration framework.

## C.4. More Super-Resolution Experimental Results

To provide a more comprehensive evaluation of the experimental performance of ASASR, we present additional visual comparisons in this section. These include qualitative results on Synthesis Datasets (Fig.9) and Real-World Datasets (Fig. 10). Furthermore, we provide visualization results for downstream high-level vision tasks, specifically illustrating OCR performance on the RoadText1K dataset (Fig. 11), object detection and instance segmentation on the COCO dataset (Fig. 12), and semantic segmentation on the ADE dataset (Fig. 13).

*Table 8.* Quantitative comparison on Bringing Old Films Back to Life. We evaluate ASASR against representative restoration and super-resolution methods on degraded old film frames using non-reference image quality assessment metrics.

| Methods | NIQE↓ | MANIQA↑ | MUSIQ↑ | CLIPIQA+↑ |
|---|---|---|---|---|
| DP2O-SR | 5.42 | 0.2621 | 32.32 | 0.5195 |
| SeeSR | 5.88 | 0.1159 | 36.70 | 0.2819 |
| DreamClear | 4.97 | 0.3842 | 45.58 | 0.4271 |
| DiT4SR | 4.51 | 0.4694 | 52.67 | 0.5004 |
| **ASASR** | **4.46** | **0.5036** | **54.42** | **0.6078** |

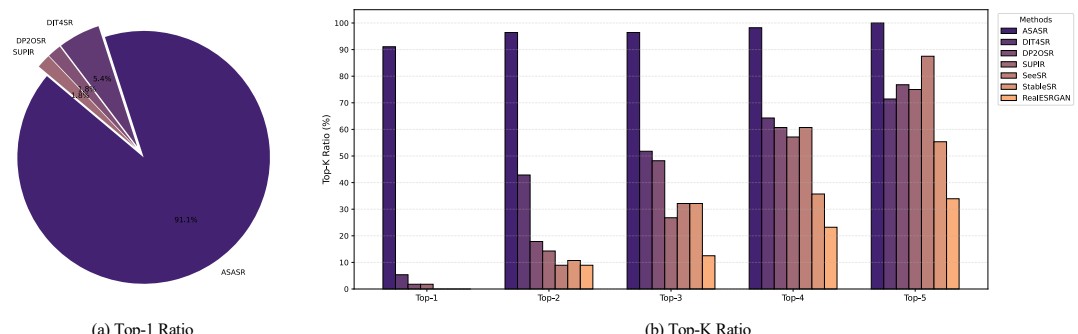

(a) Top-1 Ratio

(b) Top-K Ratio

*Figure 8.* **Visualization of user study results. (a)** Top-1 selection ratio, where ASASR secures a dominant 91.1% of user votes. **(b)** Top-K cumulative rankings, demonstrating that our method is consistently favored as the highest-quality restoration among competing baselines across all K levels.

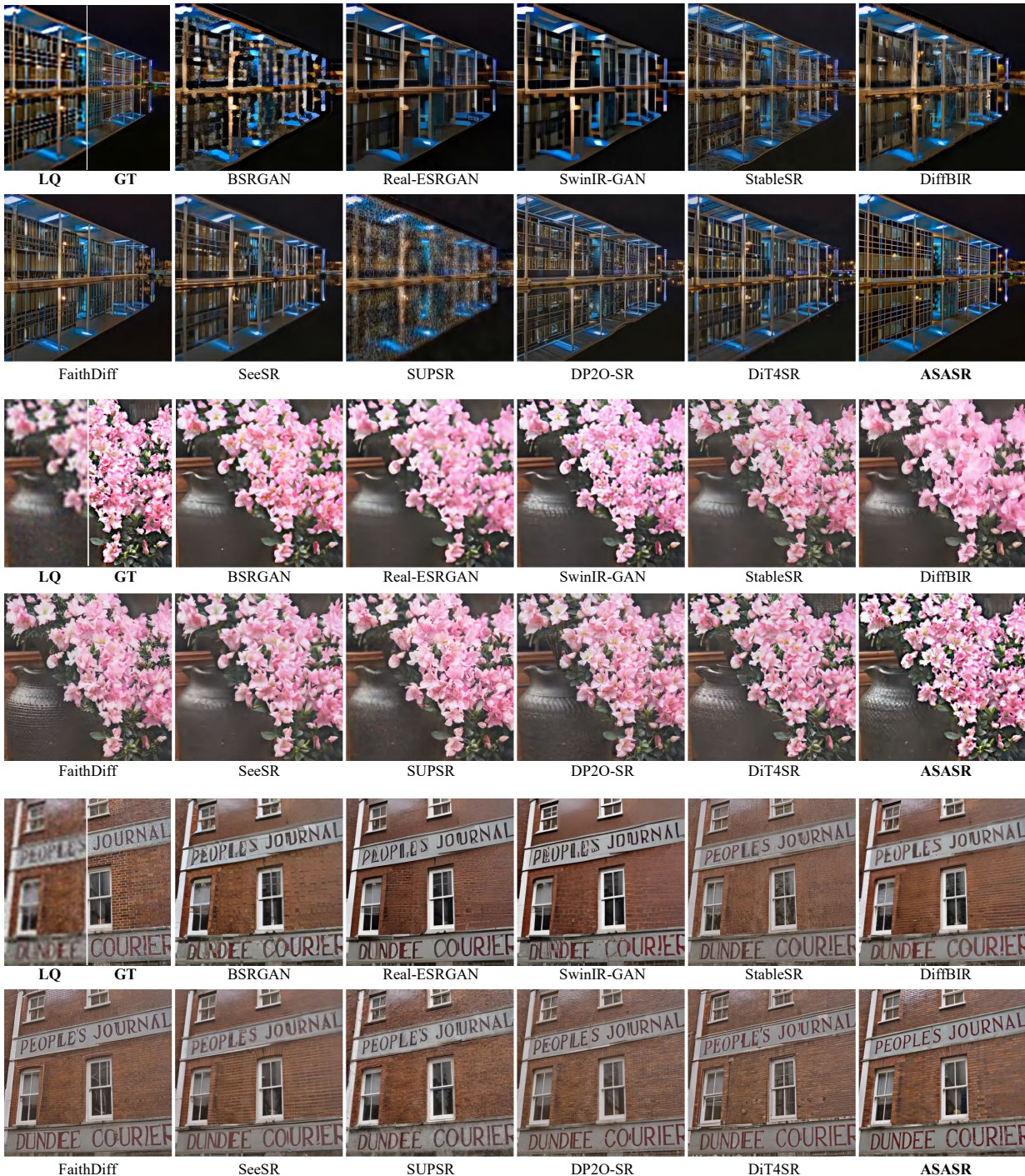

*Figure 9.* **Visual comparisons on synthesis datasets.** We compare ASASR against state-of-the-art GAN-based and diffusion-based methods. As observed, our method achieves superior structural fidelity, effectively reconstructing complex architectural geometries (1st row) and legible text (3rd row) while maintaining natural textures (2nd row), avoiding the structural distortions and hallucinations common in competing generative priors.

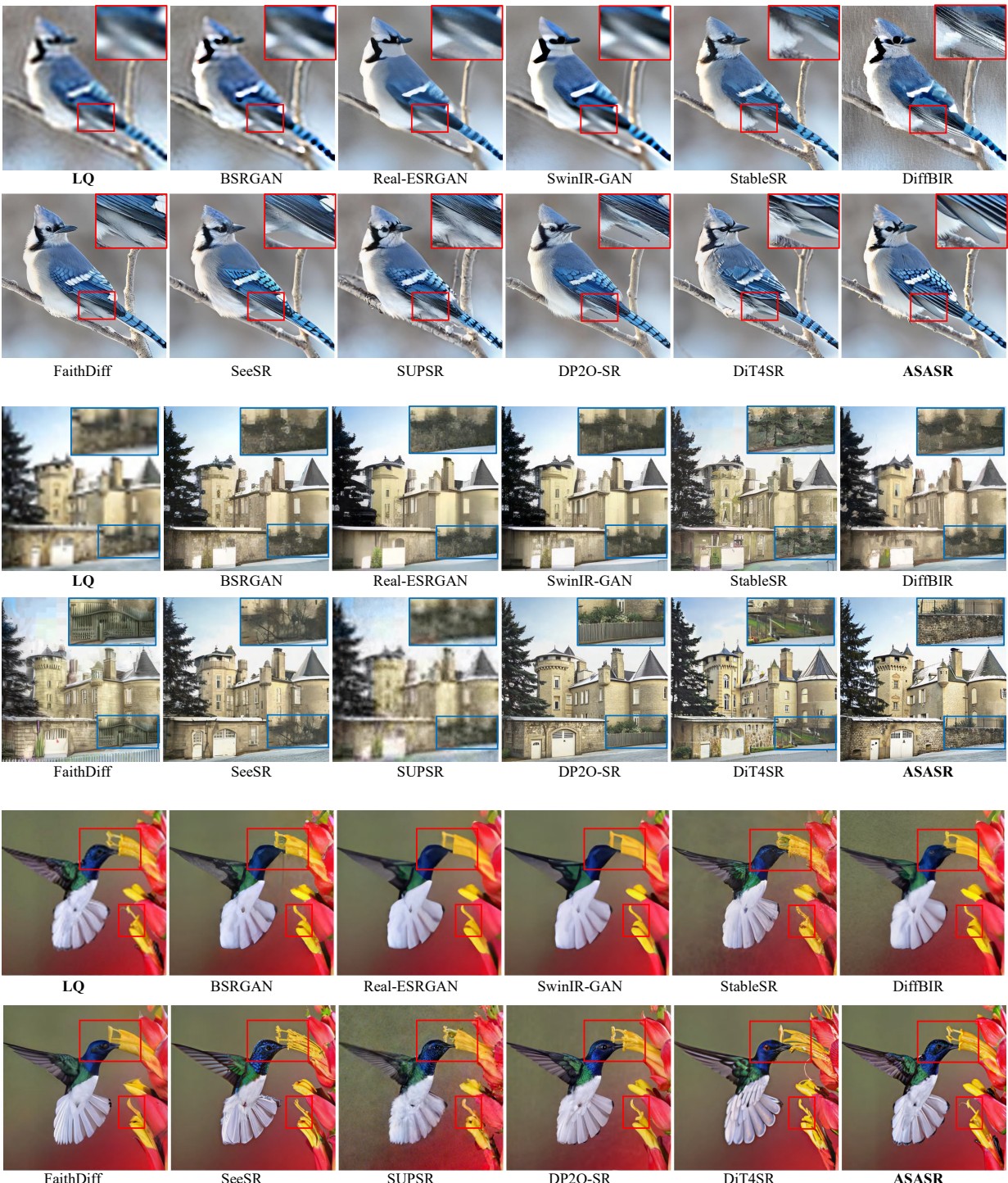

*Figure 10.* **Visual comparisons on real-world datasets.** ASASR demonstrates robust generalization capabilities on challenging real-world scenes. Unlike baselines that often produce over-smoothed textures (e.g., SwinIR) or hallucinated artifacts (e.g., StableSR), our method successfully restores intricate high-frequency details, such as feather textures (1st & 3rd rows) and distant architectural features (2nd row), striking an optimal balance between noise suppression and perceptual photorealism.

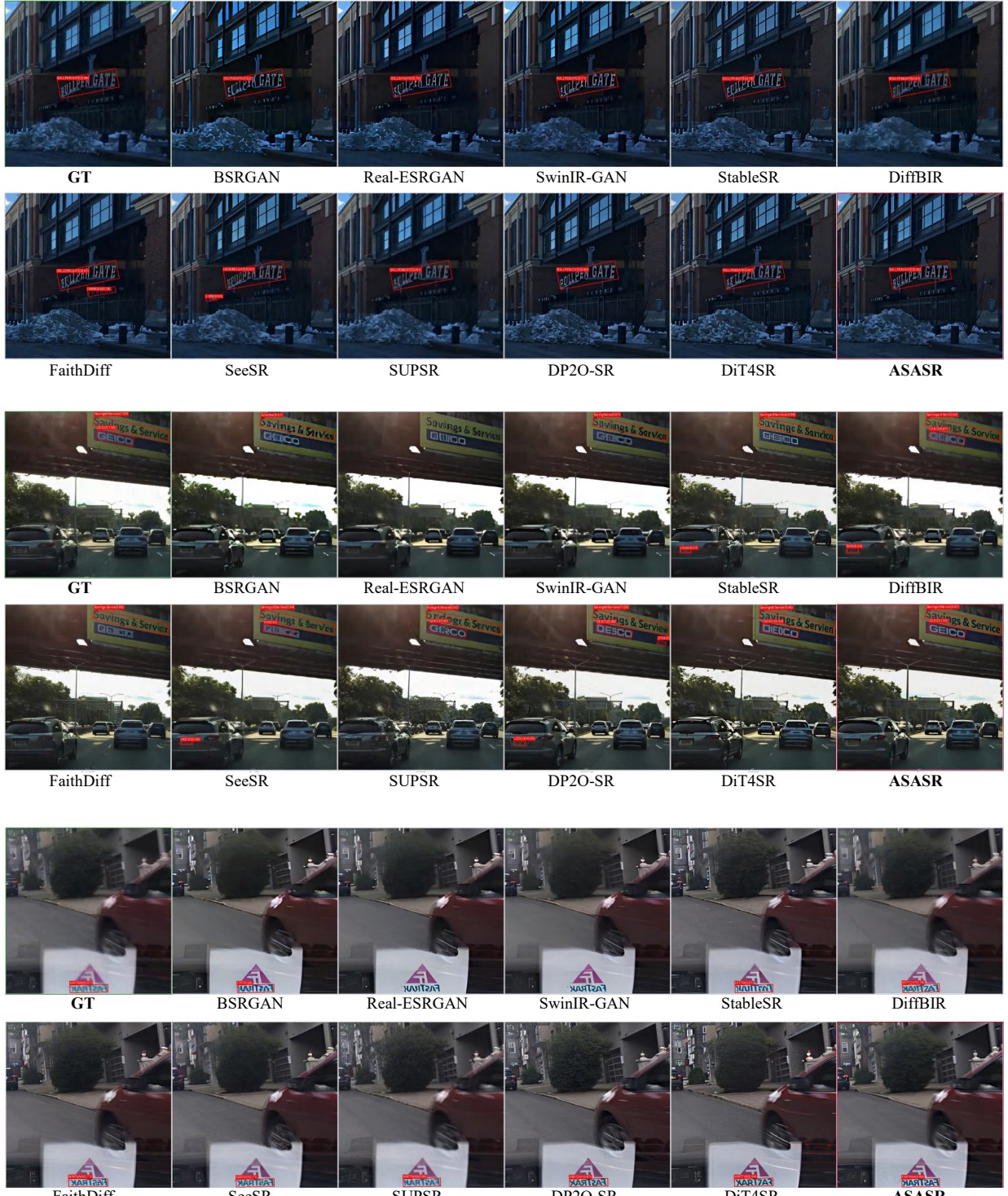

*Figure 11.* **Visualization of OCR results on the RoadText1K dataset.** To evaluate semantic preservation, we apply a pre-trained text detector on images restored by different methods. ASASR reconstructs clearer, sharper text characters compared to other generative models, enabling more accurate text detection (red bounding boxes) that closely aligns with the Ground Truth, whereas competing methods often lead to missed detections or false positives due to character corruption.

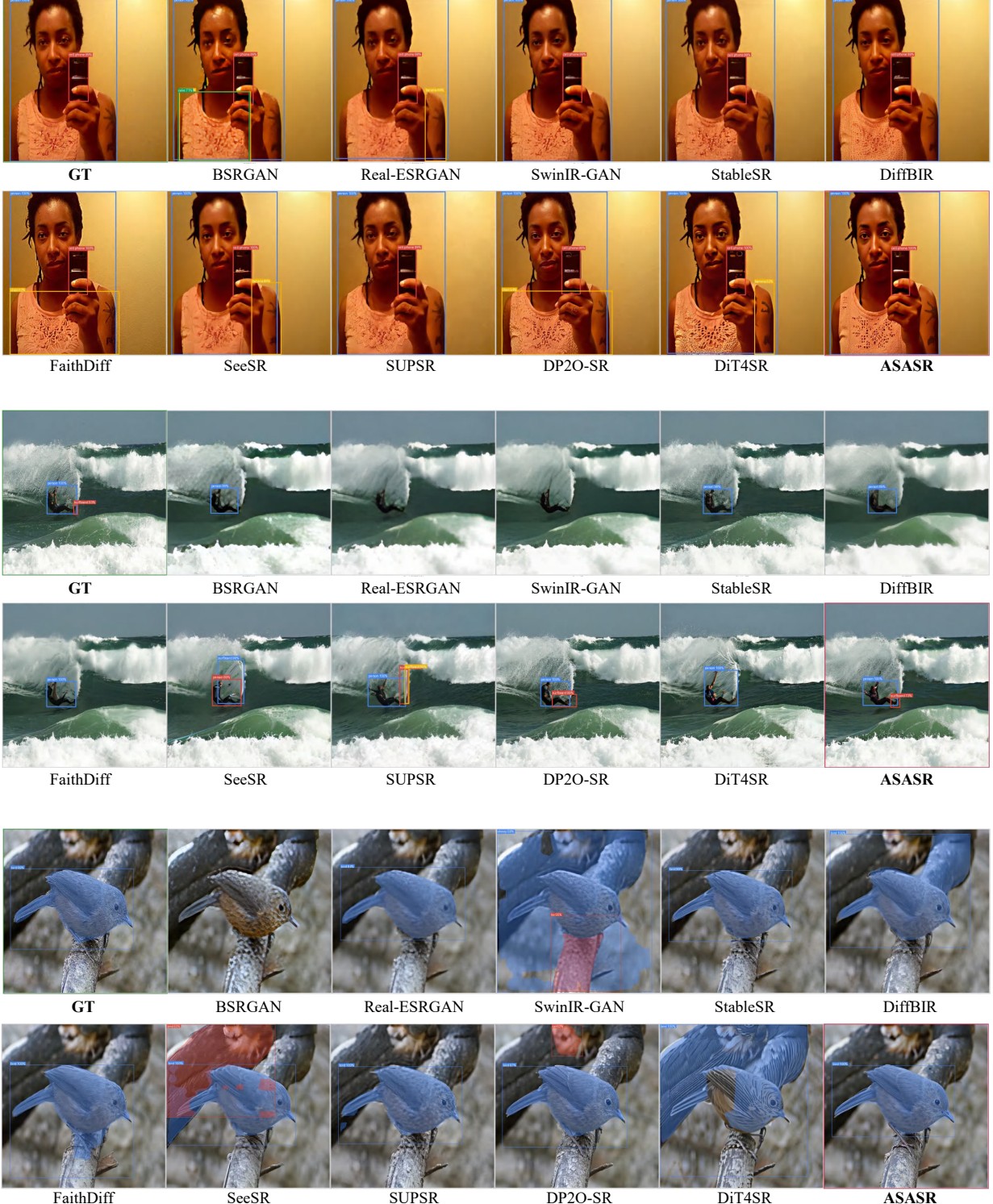

*Figure 12.* **Visualization of object detection and instance segmentation on the COCO dataset.** We visualize the detection bounding boxes and segmentation masks predicted on restored images. ASASR preserves the structural integrity of objects, such as human limbs (1st & 2nd rows) and animal boundaries (3rd row), resulting in more precise segmentation masks and higher confidence scores compared to baselines that suffer from semantic drift or boundary blurring.

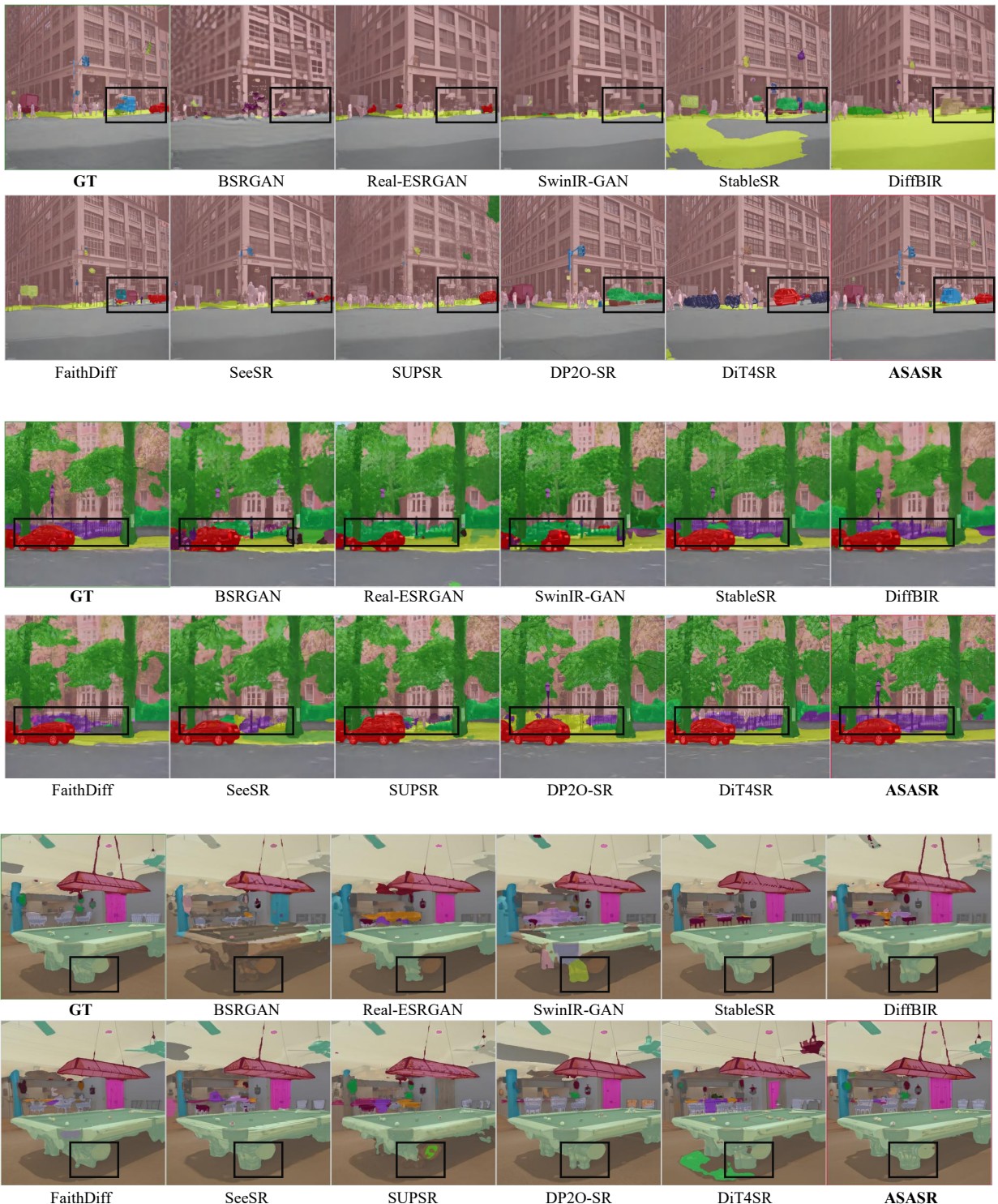

*Figure 13.* **Visualization of semantic segmentation on the ADE20K dataset.** The results illustrate the impact of restoration quality on scene parsing. ASASR effectively recovers distinct object boundaries and consistent semantic regions (e.g., the car in the 2nd row and furniture in the 3rd row), leading to cleaner segmentation maps with fewer artifacts compared to other diffusion-based counterparts, which often introduce noise that confuses the segmentation model.

