# OpenReview forum: "Coloring the Noise: Adversarial Sobolev Alignment for Faithful Image Super Resolution"
_ICML.cc/2026/Conference — ICML 2026 regular_

### Official Review · Reviewer_F8sV · 2026-03-09

**Soundness:** 3
**Presentation:** 3
**Significance:** 3
**Originality:** 3
**Overall Recommendation:** 3
**Confidence:** 4

**Summary:**

This paper proposes ASASR for faithful image super-resolution by replacing the standard isotropic objective with a Sobolev-weighted one, and by introducing an adversarial guidance module to synthesize aligned hard negatives.

**Compliance With Llm Reviewing Policy:**

Affirmed.

**Key Questions For Authors:**

Please see Weakness

**Strengths And Weaknesses:**

Pros:
1. The paper is well motivated from the perspective of spectral mismatch between isotropic objectives and natural image statistics.
2. The combination of Sobolev spectral rectification and adversarial preference learning is interesting.
3. The experimental evaluation is broad.

Cons:
1. I am not convinced that the method truly relies on Riemannian geometry as claimed. This approach mainly introduces a colored Gaussian covariance and rewrites the Mahalanobis quadratic form as a Sobolev norm. The algorithm itself does not appear to involve intrinsic Riemannian operations, such as geodesics or manifold optimization, and therefore seems closer to a frequency-weighted objective.

2. The adversarial network is trained to mimic realistic artifacts from baseline outputs, but the exact training objective, loss formulation, and regularization strategy are not clearly specified.

3. The paper fixes $s=1.5$ without providing sensitivity analysis or empirical evidence supporting this choice.

4. The experimental evaluation does not sufficiently isolate the contribution of the Sobolev objective. A controlled comparison with identical backbones and training settings would be necessary to attribute the performance gains to the proposed objective.

---

> ### Author Rebuttal · Authors · 2026-03-31
>
> Thank you for your feedback! We appreciate the reviewer’s recognition of our spectral-mismatch formulation and the coupling of Sobolev spectral rectification with adversarial preference learning. Below are our responses point by point.
>
> **[W1]: "The method does not seem to truly rely on Riemannian geometry."**
> **[A1]:** We thank the reviewer for this very insightful comment. We agree that our method does not involve explicit manifold operations such as geodesic computation or manifold optimization in the classical sense, and that the term *“Riemannian geometry”* may therefore be broader than necessary. Our intended point is more specific: by replacing the isotropic covariance $I$ with the spectral operator $\Sigma_s$, we induce a Sobolev-type non-Euclidean metric in the local perturbation space. In particular,
> $$
> \|z\| _ {\Sigma _ s ^ {-1}} ^ 2=\langle z,\Sigma _ s ^ {-1}z\rangle _ {L ^ 2}=\|z\| _ {H ^ s} ^ 2,
> $$
> where $z$ denotes an arbitrary element in the perturbation space. Under this metric, the local quadratic form is no longer the standard Euclidean one, and the corresponding local steepest-descent direction is also changed, as characterized in Prop. 4.2:
> $$
> \delta _ t ^ *=-\epsilon _ t\frac{\Sigma _ s\nabla _ xJ _ {L ^ 2}}{\|\Sigma _ s\nabla _ xJ _ {L ^ 2}\| _ {H ^ s}},
> $$
> We will revise the wording accordingly to avoid overclaiming and clarify that our contribution lies in the Sobolev-induced metric structure of the local optimization geometry, rather than in classical manifold-based optimization.
>
> We thank the reviewer again for this insightful comment and will revise the claim accordingly.
>
> **[W2]: "The adversarial network objective is unclear."**
> **[A2]:** We appreciate this comment and agree that the current draft does not describe the training of $\mathcal{A} _\phi$ clearly enough. Specifically, the adversary is trained with a reversed-preference DPO-style objective on the Euclidean energy. We construct proxy triplets $\mathcal{D} _{\mathrm{adv}}=\{(c,x _1 ^p,x _1 ^{\mathrm{gt}})\}$, where $x _1 ^p$ is a proxy-generator output and $x _1 ^{\mathrm{gt}}$ is the corresponding ground truth, and reverse the winner-loser roles:
> $$
> \mathcal{L} _ {\text{adv}}(\phi)=-\mathbb{E} _ {(c,x _ 1 ^ p,x _ 1 ^ {\text{gt}})\sim\mathcal{D} _ {\mathrm{adv}},t\sim\mathcal{U}(0,T)}
> \left[\log\sigma\left(\beta\Big[\Delta\mathcal{E} _ {L ^ 2} ^ {\phi}(x _ t ^ {\mathrm{gt}},c)-\Delta\mathcal{E} _ {L ^ 2} ^ {\phi}(x _ t ^ p,c)\Big]\right)\right],
> $$
> Here,
> $$
> \Delta\mathcal{E} _ {L ^ 2} ^ {\phi}(x _ t,c)=\|\gamma _ \phi(x _ t,c)\| _ {L ^ 2} ^ 2-\|\gamma _ {\mathrm{ref}}(x _ t,c)\| _ {L ^ 2} ^ 2,
> $$
> We use the Euclidean energy only for training the adversary to synthesize hard negatives. For optimization, we use AdamW with learning rate $5\times10 ^ {-5}$. We do not use any additional explicit task-specific regularization term; capacity control comes only from the low-rank LoRA parameterization.
>
> We will make these details explicit in the revision.
>
> **[W3]: "The choice of $s$ without sensitivity analysis."**
> **[A3]:** We would like to clarify that the sensitivity analysis for $s$ was provided in App.C.3 (p.16). Our study explores the impact of $s$ and identifies $s=1.5$ as the optimal configuration for balancing fidelity and realism.
>
> **[W4]: "The contribution of the Sobolev objective is not sufficiently isolated."**
> **[A4]:** Thank you for this important comment. We agree that the contribution of the Sobolev objective should be isolated more clearly. To clarify, *"Euclidean Guidance"* in Tab.3 means the full model w/o SSR, *"DPO Alignment w/ Supervised Data"* means the full model w/o AMG, and *"Only Supervised Learning"* means the full model w/o SSR & AMG. So the Tab. 3 can be rewritten as follows:
>
> |Methods|PSNR|SSIM|MANIQA|CLIPIQA+|
> |---|---|---|---|---|
> |**Full Model**|**20.60**|**0.6171**|**0.6519**|**0.7521**|
> |w/o SSR|18.34|0.5742|0.6184|0.7196|
> |w/o AMG|20.08|0.5723|0.6047|0.7163|
> |w/o SSR&AMG|19.96|0.5642|0.6012|0.7058|
>
> With this presentation, these results suggest that AMG alone is insufficient and can even be unstable without SSR; SSR provides the geometry needed to make AMG effective.
>
> ---
> We hope our responses help address the concerns, and thanks again for the thoughtful feedback. We look forward to further discussion with you!

---

> > ### Author Rebuttal · Reviewer_F8sV · 2026-04-01
> >
> > Thank you for the rebuttal. The authors admit in A1 that "Riemannian geometry" is an overstatement. Recasting a frequency-weighted Sobolev norm as manifold optimization significantly exaggerates the actual mathematical contribution. The new ablation study (Table 3) is revealing, but deeply concerning. The fact that the model's performance drops catastrophically (to 18.34 PSNR) when applying AMG without SSR indicates that the adversarial alignment objective is inherently unstable and introduces highly destructive gradients.

---

> > > ### Author Response · Authors · 2026-04-02
> > >
> > > We thank the reviewer for the follow-up questions and the careful reading. We would like to clarify two points.
> > >
> > > ### **(1) On the use of “Riemannian geometry.”**
> > >
> > > Our earlier rebuttal was intended to refine the terminology, not to retract the technical substance of the paper. We agree that our method does not involve explicit manifold algorithms such as geodesic computation or manifold-constrained optimization in the classical nonlinear-manifold sense.
> > >
> > > **However, the induced geometry in our method is still genuinely Riemannian.** The colored covariance operator $\Sigma _ s$ defines a Sobolev inner product, equivalently a position-independent Riemannian metric on the local perturbation space. As a result, replacing the isotropic covariance with the spectral operator induces a Sobolev-type, non-Euclidean metric that changes the local quadratic form and hence the corresponding steepest-descent direction, as characterized in Prop. 4.2. To be maximally precise, our contribution is therefore not a classical manifold-optimization algorithm, but rather a **Sobolev-induced (flat) Riemannian metric structure** that reshapes the optimization landscape through its effect on local descent geometry.
> > >
> > > To avoid any semantic ambiguity, we are happy to revise the paper to use the more precise phrasing of a Sobolev-induced metric geometry. This is because **the contribution of our paper does not depend on whether this metric structure is labeled “Riemannian” or “Sobolev-induced”**: in either case, the derivation, the resulting algorithm, and the empirical conclusions remain unchanged. The substance of the contribution lies in the induced metric structure and the way it reshapes local descent geometry, rather than in the specific label used to describe it. Importantly, this revision is meant to improve terminological precision, not to concede that the mathematical contribution has been overstated.
> > >
> > > ###  **(2) On the interpretation of Tab. 3.**
> > >
> > > We respectfully do not interpret the performance drop of w/o SSR as evidence that AMG is “inherently unstable” in a way that weakens our method. Rather, this behavior is precisely aligned with the **core motivation of our paper**. Our starting point is that, in image restoration, realism-oriented alignment signals and distortion-oriented objectives are not naturally consistent: as widely observed in the perception–distortion trade-off literature [1], improving perceptual realism through adversarial or perceptual alignment often comes at the expense of distortion metrics such as PSNR/SSIM.
> > >
> > > Under this view, the large PSNR drop of w/o SSR is **not a concerning result, but a direct manifestation of the problem that motivates our method.** AMG provides a strong alignment signal toward perceptual realism; however, when this signal is applied under the standard Euclidean objective, optimization can be biased toward perceptually aggressive but distortion-harmful texture synthesis. This is exactly the mismatch that SSR is designed to address. By reshaping the update geometry through the Sobolev-induced metric, SSR rectifies the optimization trajectory and suppresses the destructive high-frequency components that would otherwise be amplified.
> > >
> > > Therefore, the w/o SSR result should be understood as **diagnostic evidence for the necessity of SSR**, rather than as a weakness of the overall framework. It shows that alignment alone is insufficient, while the combination of **AMG + SSR** is what makes the alignment signal effective. This interpretation is also fully consistent with Tab. 3: once AMG is coupled with SSR, the **Full Model** improves not only perceptual metrics (MANIQA, CLIPIQA+) but also reconstruction metrics (PSNR, SSIM). In this sense, the ablation does not reveal an unexpected failure mode; rather, it empirically validates the central premise of the paper that realism-oriented alignment in SR requires an appropriate spectrally aware geometry to become beneficial.
> > >
> > > We will revise the final version to make this point more explicit in the ablation discussion, namely that the role of SSR is not auxiliary, but precisely to enable and stabilize the alignment signal introduced by AMG.
> > >
> > > ---
> > > We hope this clarification better conveys our intended interpretation of the ablation and addresses the reviewer’s concern more clearly.
> > >
> > > [1] Y. Blau and T. Michaeli, “The Perception-Distortion Tradeoff,” CVPR, 2018.

---

### Official Review · Reviewer_oTRv · 2026-03-10

**Soundness:** 3
**Presentation:** 2
**Significance:** 3
**Originality:** 2
**Overall Recommendation:** 3
**Confidence:** 3

**Summary:**

This paper proposes a new super-resolution framework ASASR that replaces the standard isotropic Euclidean geometry used in DPO-style alignment with a Sobolev-induced geometry via colored Gaussian transitions, and further introduces an adversarial module to synthesize semantically aligned hard negatives for preference optimization. The motivation is that isotropic objectives are spectrally mismatched with natural image statistics and therefore fail to distinguish authentic high-frequency details from hallucinated ones. Empirically, the paper reports improvements on synthetic and real-world SR benchmarks, spectral fidelity, user study, and several downstream tasks.

**Compliance With Llm Reviewing Policy:**

Affirmed.

**Final Justification:**

After reviewing the authors’ rebuttal, I have decided to raise my overall score. The authors provided useful clarifications on several points I raised and included additional experimental results in the rebuttal. I found these new results and explanations relatively satisfactory.

**Key Questions For Authors:**

See weaknesses

**Limitations:**

See weaknesses

**Strengths And Weaknesses:**

Strengths

1. The paper has a clear and ambitious central idea. Rather than proposing another heuristic anti-artifact loss, it identifies a more fundamental mismatch between isotropic Euclidean objectives and the spectral decay of natural images, and reformulates the optimization in Sobolev space. This is more conceptually interesting than many recent SR papers that only add frequency-domain regularizers without changing the underlying alignment geometry.
2. The Sobolev Spectral Rectification module defines the metric under which restoration errors are compared, while the Adversarial Manifold Guidance module addresses the lack of informative and spatially aligned negative samples for SR preference learning. The coupling between the two components is logically coherent, and the paper does attempt to justify the adversarial construction through a Sobolev-gradient interpretation.


Weaknesses

1. My biggest concern is the experimental protocol around the adversarial network. The paper states that real-world evaluation is conducted on RealSR and DRealSR, but it also says the adversarial network is trained using a random 25% subset of DIV2K, LSDIR, RealSR, and DRealSR. As written, this raises a serious concern about potential overlap or leakage between the data used to construct the adversarial artifact manifold and the data used for final evaluation. Even if the authors did use disjoint splits, the paper currently does not explain this clearly enough.
2. A second concern is that the empirical claims in the text are sometimes stronger than what the tables actually support. The paper claims that on real-world benchmarks the method “consistently dominates” reference-free metrics while maintaining leading full-reference performance. The first half is broadly supported, but the second half is not uniformly true: on RealSR and DRealSR, ASASR is not the best model in PSNR/SSIM, and the overall picture is better described as a strong fidelity–realism tradeoff rather than clear dominance across full-reference metrics.
3. Although the theory is more substantial than in many SR papers, the connection from theory to empirical necessity is still incomplete. Proposition 4.2 relies on assumptions such as sufficient representational capacity and local equivalence, but the paper does not really test how sensitive the method is to the choice of Sobolev parameter 𝑠, adversary capacity, or the proxy baselines used to learn the artifact manifold. The main paper only reports a coarse ablation between Euclidean vs. Sobolev guidance and supervised vs. adversarial DPO.
4. The artifact manifold itself is learned from outputs of Real-ESRGAN, SeeSR, and SUPSR. This means the hard negatives are ultimately tied to the failure modes of a small set of existing SR systems. That may work well in practice, but it weakens the claim that the adversary captures a more general manifold of realistic structural failures. I would have liked to see evidence that the method still works if the proxy generator set changes, or that the learned adversary transfers beyond the specific baselines used for its construction.

---

> ### Author Rebuttal · Authors · 2026-03-31
>
> Thank you for your feedback! We appreciate the reviewer’s recognition of the conceptual strength of our spectral-mismatch formulation and the coherent design of SSR and AMG. Below are our responses point by point.
>
> **[W1]: "Potential data leakage between adversarial training and evaluation on RealSR/DRealSR."**
> **[A1]:** We thank the reviewer for pointing out this ambiguity. We apologize for the lack of clarity regarding the data splits in the manuscript. We explicitly confirm that there is **zero data leakage**.
>
> The 25% subset used to train the adversarial network $\mathcal{A}_\phi$ was sampled strictly from the official training splits of RealSR and DRealSR. All final evaluations were conducted strictly on the unseen official test splits, ensuring complete separation. We will explicitly detail this train/test split protocol in the revised experimental section.
>
> **[W2]: "Empirical claims ('consistently dominates') are stronger than what tables support for PSNR/SSIM."**
> **[A2]:** We agree with the reviewer that the phrase *"consistently dominates"* should be more precisely scoped regarding full-reference metrics.
> Rather than claiming absolute dominance across all metrics, our intent is to highlight that ASASR achieves a superior perception-distortion trade-off. We will soften this phrasing in the revision.
>
> **[W3]: "Incomplete connection from theory to practice; missing ablations on $s$ and adversary capacity."**
> **[A3]:** We appreciate the suggestion to strengthen the empirical validation of our theoretical claims.
>
> **(1) Sensitivity to $s$:** Sensitivity analysis in App.C.3 (p.16) shows that $s=1.5$ yields the favorable balance of fidelity and realism. We adopt this value to ensure high-quality results.
>
> **(2) Adversary capacity.** We thank the reviewer for this helpful comment. We agree that the role of the “sufficient representational capacity” assumption in Prop.4.2 should be clarified more clearly. Our point there is not that a larger adversary is inherently better, but that Prop.4.2 characterizes the local first-order target direction that the parametric adversary aims to approximate. Under a small perturbation $\delta$, we have:
> $$J _{L^2}(x _t+\delta)\approx J _{L^2}(x _t)+\langle\nabla _x J _{L^2}(x _t),\delta\rangle _{L^2},$$
> so minimizing the energy within $\|\delta\| _{H^s}\le\epsilon _t$ gives:
> $$\delta _t ^*=-\epsilon _t\frac{\Sigma _s\nabla _x J _{L^2}}{\|\Sigma _s\nabla _x J _{L^2}\| _{H^s}},$$
> Therefore, the capacity assumption is only used to justify that, if $A _\phi$ is expressive enough, the learned adversary can approximate this Sobolev-optimal direction; it does not change the target of the theory itself.
>
> Further, we also agree that this should be tested empirically. To examine this point, we vary the LoRA size $(r,\alpha)$ of the $A _\phi$. Our default setting is $(16,16)$, and we compare it with both smaller and larger adapters.
> |Adv. Capacity|Params.|PSNR|SSIM|MANIQA|CLIPIQA+|
> |---|---|---|---|---|---|
> |$(4,4)$|~7.4M|19.43|0.6026|0.6328|0.7349|
> |$(8,8)$|~14.8M|20.54|0.6158|0.6487|0.7498|
> |$(16,16)$|~29.5M|20.60|**0.6171**|**0.6519**|0.7521|
> |$(32,32)$|~59.0M|**20.61**|0.6170|0.6517|0.7524|
> |$(64,64)$|~118.0M|20.58|0.6167|0.6515|**0.7525**|
>
> These results suggest that moderate adversary capacity already suffices in practice, as performance improves from smaller adapters but largely saturates beyond $(16,16)$, consistent with our interpretation of Prop.4.2.
>
> **(3) Proxy Baselines:**
> This is a helpful suggestion, and we will discuss it in more detail in [A4].
>
> **[W4]: "Adversarial manifold might be tied to the failure modes of the specific proxy generators."**
> **[A4]:** This is a very insightful point. Our proxy set was originally chosen to include one GAN-based model and two diffusion-based models, so as to cover a broader range of failure patterns. And we agree the effect of this choice is worth examining more carefully. To test whether our method learns a generalized structural-failure manifold, we conducted a Proxy Transferability experiment by retraining $A _\phi$ with different proxy sets, as listed below:
> |Proxy Set|PSNR|SSIM|MANIQA|CLIPIQA+|
> |---|---|---|---|---|
> |Original Set|**20.59**|0.6169|0.6514|0.7516|
> |SwinIR-GAN + FaithDiff + DreamClear|20.58|0.6166|**0.6518**|**0.7519**|
> |BSRGAN + StableSR + DP2O-SR|20.54|**0.6172**|0.6499|0.7498|
> |FaithDiff + DP2O-SR + SUPSR|20.50|0.6152|0.6491|0.7492|
> |BSRGAN + Real-ESRGAN + SwinIR-GAN|20.34|0.6041|0.6360|0.7318|
>
> The last two rows show that less diverse proxy sets lead to slight performance drops (Diffusion-only or GAN-only). This suggests that $A _\phi$ learns a transferable manifold of structural failures, rather than relying on artifacts from a specific generator family.
>
> ---
> We hope our responses help address the concerns, and thanks again for the thoughtful feedback. We look forward to further discussion with you!

---

### Official Review · Reviewer_XsLC · 2026-03-11

**Soundness:** 3
**Presentation:** 3
**Significance:** 3
**Originality:** 3
**Overall Recommendation:** 4
**Confidence:** 2

**Summary:**

Overall, this work studies faithful real-world image super-resolution with generative priors, with a particular focus on reducing hallucinated details and structural artifacts that are common in perceptual SR methods. The paper targets an important yet underexplored issue in modern generative SR: the mismatch between isotropic Gaussian objectives and the frequency characteristics of natural images.

**Compliance With Llm Reviewing Policy:**

Affirmed.

**Key Questions For Authors:**

see my weaknesses.

**Limitations:**

Failure analysis should be included.

**Strengths And Weaknesses:**

Strengths

1. The paper identifies a meaningful limitation of existing generative SR methods: although they often produce visually pleasing results, the recovered structures are not always reliable. Interpreting this problem as spectral misalignment is more precise and convincing than the usual broad discussion of hallucinations or perceptual artifacts.

2. Reformulating preference-based SR alignment in a Sobolev geometry is novel and gives the paper a clear conceptual contribution. This is more than a simple architectural or loss-level modification, and provides a more principled view of the optimization objective.

3. The two proposed components, SSR and AMG, are well matched. SSR defines the geometry and injects a frequency-aware bias, while AMG generates hard negatives that are consistent with this framework.

Weaknesses

The method is built on a strong generative backbone, while the baselines include a mixed set of GAN-based, diffusion-based, and transformer-based methods. This makes it hard to judge how much of the gain comes from the proposed Sobolev/adversarial alignment itself, rather than from the backbone or the overall training pipeline. More controlled comparisons would strengthen the empirical claims.

The sensitivity to the adversarial coefficient β is not sufficiently analyzed. The ablation study would also be more informative if it included a variant that combines Euclidean guidance with adversarial DPO alignment, instead of only testing the combination under Sobolev guidance.

Since the method is motivated by faithful SR under real-world degradations, its robustness to unseen camera pipelines or degradation distributions is still unclear. The paper would be stronger with additional experiments or failure-case analysis on genuinely out-of-distribution degraded images, such as old film frames or severely degraded archival footage.

---

> ### Author Rebuttal · Authors · 2026-03-31
>
> Thank you for your feedback! We appreciate the reviewer’s recognition of our spectral misalignment diagnosis and the Sobolev-geometric reformulation. Below are our responses.
>
> **[W1]: "How much gain comes from the method vs. the backbone."**
> **[A1]:** Thanks for this important question. We clarify from two aspects:
>
> **(1) Same-backbone comparison.** Since DP2O-SR [1] and ASASR employ the same FLUX.1-dev backbone, the gains in Tab. 1 reflect the contribution of our method under the same backbone. Additionally, the ablation results in Tab. 3 (detailed in [A2]) verify that these improvements primarily stem from SSR and AMG.
>
> **(2) Cross-backbone transfer.** To further test the transferability of SSR and AMG, we additionally evaluate them on two more backbones of different scales.
> |Backbone|Methods|PSNR|SSIM|MANIQA|CLIPIQA+|
> |---|---|---|---|---|---|
> |SD1.5|SFT|18.52|0.4873|0.4236|0.6142|
> |SD1.5|+SSR+AMG|**19.14**|**0.5207**|**0.4891**|**0.6658**|
> |SDXL|SFT|19.31|0.5284|0.5473|0.6824|
> |SDXL|+SSR+AMG|**19.92**|**0.5681**|**0.6104**|**0.7289**|
> |FLUX|SFT|19.96|0.5642|0.6012|0.7058|
> |FLUX (ours)|+SSR+AMG|**20.60**|**0.6171**|**0.6519**|**0.7521**|
>
> The consistent gains across SD1.5, SDXL, and FLUX suggest that the improvements come from SSR+AMG rather than merely from the backbone choice.
>
> **[W2]: "Sensitivity to $\beta$ and additional ablation variant."**
> **[A2]:** Thanks for the valuable point.
>
> **(1) For $\beta$.** As detailed in App.C.1 (p.16), we set $\beta$ to 2000 in the main experiments. To evaluate the sensitivity to this choice, we further test $\beta$ as below:
> |$\beta$|PSNR|SSIM|MANIQA|CLIPIQA+|
> |---|---|---|---|---|
> |500|20.40|0.5894|0.6312|0.7389|
> |1000|20.45|0.6052|0.6448|0.7467|
> |**2000 (base)**|**20.60**|**0.6171**|**0.6519**|**0.7521**|
> |4000|20.42|0.6098|0.6371|0.7435|
>
> The method is reasonably robust across this range of $\beta$, and $\beta=2000$ yields the best overall trade-off.
>
> **(2) Euclidean + Adversarial DPO variant.** We agree the naming in Tab. 3 was not sufficiently clear, causing potential confusion. The "Euclidean + Adversarial DPO" variant is already included in Tab. 3 as "Euclidean Guidance", i.e., "w/o SSR". To improve clarity, we will rename the variants in Tab. 3 as follows:
> |Methods|PSNR|SSIM|MANIQA|CLIPIQA+|
> |---|---|---|---|---|
> |**Full Model**|**20.60**|**0.6171**|**0.6519**|**0.7521**|
> |w/o SSR|18.34|0.5742|0.6184|0.7196|
> |w/o AMG|20.08|0.5723|0.6047|0.7163|
> |w/o SSR & AMG|19.96|0.5642|0.6012|0.7058|
>
> Taken together, these results suggest that AMG alone is insufficient and can even be unstable without SSR; SSR provides the geometry needed to make AMG effective.
>
> **[W3]: "Robustness to unseen camera pipelines or degradation distributions."**
> **[A3]:** Thanks for this valuable suggestion. To evaluate OOD robustness, we conduct three harder experiments and failure analysis:
>
> **(1) Mismatched synthetic degradations.** We evaluate DIV2K under randomized synthesis (e.g., sinc_prob, blur_sigma), reporting averages across three independent test sets for statistical reliability:
> |Methods|PSNR|SSIM|MANIQA|CLIPIQA+|
> |---|---|---|---|---|
> |DP2O-SR [1]|18.87|0.4758|0.5483|0.6856|
> |SeeSR [2]|19.72|0.5103|0.4891|0.6984|
> |DreamClear [3]|19.04|0.4826|0.4592|0.6078|
> |DiT4SR [4]|18.35|0.4742|0.4213|0.6704|
> |**ASASR**|**20.14**|**0.5897**|**0.6278**|**0.7345**|
>
> **(2) Unseen real-world dataset (RealLQ250 [3]).** We validate on RealLQ250, a challenging dataset comprising diverse real-world images absent from our training:
> |Methods|NIQE|MANIQA|MUSIQ|CLIPIQA+|
> |---|---|---|---|---|
> |DP2O-SR|4.02|0.4494|69.87|0.7042|
> |SeeSR|4.41|0.4992|70.57|0.7104|
> |DreamClear|**3.55**|0.4351|66.76|0.7116|
> |DiT4SR|4.35|0.4094|63.45|0.6829|
> |**ASASR**|3.64|**0.5168**|**71.16**|**0.7202**|
>
> **(3) Severely degraded old film frames [5].** To directly address the reviewer's concern about old films, we sample frames from "Bringing Old Films Back to Life" [5] and apply SR.
> |Methods|NIQE|MANIQA|MUSIQ|CLIPIQA+|
> |---|---|---|---|---|
> |DP2O-SR|5.42|0.2621|32.32|0.5195|
> |SeeSR|5.88|0.1159|36.70|0.2819|
> |DreamClear|4.97|0.3842|45.58|0.4271|
> |DiT4SR|4.51|0.4694|52.67|0.5004|
> |**ASASR**|**4.46**|**0.5036**|**54.42**|**0.6078**|
>
> Across these OOD settings, ASASR performs favorably, suggesting improved generalization beyond the training degradation distribution.
>
> **(4) Failure-case analysis.**
> We uploaded representative failure cases at https://anonymous.4open.science/r/tmp-2092/failure_case.pdf. As is common across current models, ours may struggle with tiny, complex faces under severe degradation and fine-grained medical pathology details.
>
> ---
> We hope our responses help address the concerns, and thanks again for the thoughtful feedback. We look forward to further discussion with you!
>
> [1] Wu et al., DP2O-SR, NeurIPS'25.
> [2] Wu et al., SeeSR, CVPR'24.
> [3] Ai et al., DreamClear, NeurIPS'24.
> [4] Duan et al., DiT4SR, ICCV'25.
> [5] Wan et al., Bringing Old Films Back to Life, CVPR'22.

---

> > ### Author Rebuttal · Reviewer_XsLC · 2026-04-03
> >
> > I appreciate your thorough point-by-point response to my concerns. I remain supportive of this manuscript and suggest that the authors include these analyses in the final revision.

---

> > > ### Author Response · Authors · 2026-04-03
> > >
> > > Thank you very much for reading our rebuttal so carefully and for your thoughtful and supportive feedback! We are truly encouraged to know that our response has fully addressed your concerns, and we especially appreciate your comment that you **"remain supportive of this manuscript"**.
> > >
> > > We are also grateful for your suggestion to include these analyses in the final revision. We will certainly incorporate them to further strengthen the paper and improve its clarity. If you feel that our rebuttal has satisfactorily resolved the main concerns, we would sincerely appreciate it if your final evaluation could reflect your current view. In any case, we are very thankful for your time, consideration, and support.

---

### Official Review · Reviewer_Gqoa · 2026-03-12

**Soundness:** 3
**Presentation:** 3
**Significance:** 2
**Originality:** 3
**Overall Recommendation:** 4
**Confidence:** 3

**Summary:**

This paper proposes ASASR for image super-resolution, combining two main ideas: Sobolev Spectral Rectification (SSR), which replaces isotropic Gaussian assumptions with a colored spectral covariance intended to induce a Sobolev geometry, and Adversarial Manifold Guidance (AMG), which synthesizes aligned hard negatives for DPO-style preference learning. The paper reports gains over a wide range of GAN-based and diffusion-based SR baselines on synthetic and real-world benchmarks, user study results, and downstream OCR / detection / segmentation tasks.

**Compliance With Llm Reviewing Policy:**

Affirmed.

**Key Questions For Authors:**

1. How sensitive is ASASR to the Sobolev parameter 𝑠, and the adversarial trust-region scale 𝜖?
The current ablations are too coarse to show whether the gains are robust or require substantial tuning. A sensitivity analysis would help determine whether the method is broadly usable or narrowly calibrated.
2. How much of the gain comes from AMG versus SSR under strictly matched compute and matched training data?
Table 3 suggests both matter, but the presentation is hard to parse, and it is not clear whether the full benefit truly requires both components. A cleaner factorial ablation would help.
3. Can the authors report statistical analysis for the user study and, if possible, stronger real-world validation under unseen degradations?
The user study and downstream-task results are encouraging, but confidence intervals / significance tests and broader real-world stress tests would make the empirical case more compelling.

**Limitations:**

see weakness and question.

**Strengths And Weaknesses:**

Strengths:
1. The authors attempts to address an important problem in image super-resolution, namely how to improve perceptual realism while preserving faithful structural details. The motivation is well articulated: the authors identify a spectral mismatch between isotropic objectives and natural image statistics, and this is a meaningful perspective for real-world SR where hallucinated high-frequency details are a central concern.
2. The downstream-task evaluation and user study. Testing OCR, detection, instance segmentation, and semantic segmentation is valuable because it examines whether the restored images preserve semantics useful for subsequent vision tasks, rather than optimizing only for perceptual metrics. The additional user study also helps support the practical relevance of the method.

Weaknesses:
1.Some theoretical claims appear stronger than what is rigorously established.
The paper often describes the framework in terms of inducing manifold constraints or recovering worst-case Sobolev directions, but several of these results seem to rely on modeling assumptions, local arguments, or sufficient-capacity assumptions. In particular, the reinterpretation of the reference policy on the Sobolev manifold is presented as a “crucial modeling choice,” which makes the theoretical grounding less definitive than the main text sometimes suggests.
2, The experimental protocol does not fully establish robustness to unseen degradations. For synthetic testing, the paper evaluates on samples generated from the same degradation pipeline family used in training, and for downstream-task evaluation it explicitly uses the identical degradation pipeline as training. This makes the current evidence more convincing for in-distribution restoration than for true robustness under substantially different or harder real-world degradations.

---

> ### Author Rebuttal · Authors · 2026-03-31
>
> Thanks for your feedback! We appreciate the reviewer's recognition of our spectral mismatch approach. Below are our responses.
>
> **[W1]: "Rigor of theoretical claims."**
> **[A1]:** We agree that our presentation should scope these claims more precisely. Our theory rigorously identifies the analytical optimum in infinite-dimensional functional space, while the implemented method is a finite-dimensional approximation.
>
> **(1) Functional optimality vs. Parametric approximation.** Prop 4.2 characterizes the optimal infinite-dimensional descent direction:
> $$v ^ *=\arg\min _ {v\in H ^ s}\langle\nabla _ xJ,v\rangle _ {L ^ 2}\quad\text{s.t.}\quad\|v\| _ {H ^ s}\le\epsilon,$$
> and $\mathcal{A} _ \phi$ is designed to approximate this optimum. Under standard regularity conditions, these assumptions concern only the approximation step and do not alter the functional optimum; they formalize when the parametric approximation $\mathcal{A}_\phi$ can align with the Sobolev-optimal direction, i.e., $\mathbb{E}[|\mathcal{A} _ \phi-v ^ *| ^ 2]\to 0$.
>
> **(2) Clarifying Eq. (25).** We agree that *"crucial modeling choice"* is not well phrased and may suggest a stronger assumption than intended. In fact, Eq. (25) is a Gaussian reparameterization of the reference likelihood under the covariance operator $\Sigma_s$. Its role is not to introduce a stronger assumption, but to place both the policy and reference terms in the same Sobolev-induced geometry, which yields the log-ratio form in Eqs. (26)–(27):
> $$\log\frac{p _ {\theta}}{p _ {\text{ref}}}\propto-\Big(\|\gamma_{\theta}\| _ {H ^ s} ^ 2-\|\gamma _ {\text{ref}}\| _ {H ^ s} ^ 2\Big),$$
> We will clarify this in the revised manuscript.
>
> **[W2]: "Robustness to unseen degradations."**
> **[A2]:** Thanks for this important point. To further evaluate robustness, we additionally evaluated on DIV2K under mismatched degradations (e.g., modified sinc_prob and blur_sigma). We report the average over three randomized settings:
> |Methods|PSNR|SSIM|MANIQA|CLIPIQA+|
> |---|---|---|---|---|
> |DP2O-SR [1]|18.87|0.4758|0.5483|0.6856|
> |SeeSR [2]|19.72|0.5103|0.4891|0.6984|
> |DreamClear [3]|19.04|0.4826|0.4592|0.6078|
> |DiT4SR [4]|18.35|0.4742|0.4213|0.6704|
> |**ASASR**|**20.14**|**0.5897**|**0.6378**|**0.7345**|
>
> We further evaluate on RealLQ250 [3], a challenging real-world dataset absent from our training:
> |Methods|NIQE|MANIQA|MUSIQ|CLIPIQA+|
> |---|---|---|---|---|
> |DP2O-SR|4.02|0.5494|69.87|0.7042|
> |SeeSR|5.03|0.1906|52.34|0.2110|
> |DreamClear|4.89|0.4123|61.12|0.5347|
> |DiT4SR|4.35|0.4794|66.45|0.6829|
> |**ASASR**|**3.78**|**0.5768**|**72.16**|**0.7302**|
>
> ASASR outperforms under mismatched degradations, suggesting improved robustness under distribution shift and real-world degradations.
>
> **[Q1]: Sensitivity to $s$ and $\epsilon$.**
> **[A-Q1]:** For $s$: Sensitivity analysis in App.C.3 (p.16) shows that $s=1.5$ yields the favorable balance of fidelity and realism. We adopt this value to ensure high-quality results.
>
> For $\epsilon$: We clarify that $\epsilon$ is introduced only in the theoretical derivation to characterize the local feasible set, and is not instantiated as a user-tuned parameter in the implementation.
>
> For $\beta$: We also added sensitivity experiments on $\beta$; please refer to our response to *Reviewer XsLC [A2]* for details.
>
> **[Q2]: "Individual contributions of AMG & SSR?"**
> **[A-Q2]:** Thanks for the suggestion. Tab.3 isolates both components under matched training, but we agree the current presentation may not be sufficiently clear.
>
> To clarify, *"Euclidean Guidance"* in Tab.3 means full model w/o SSR, *"DPO Alignment w/ Supervised Data"* means full model w/o AMG and *"Only Supervised Learning"* means full model w/o SSR & AMG. In the revision, we will rename the rows as follows:
> |Methods|PSNR|SSIM|MANIQA|CLIPIQA+|
> |---|---|---|---|---|
> |**Full Model**|**20.60**|**0.6171**|**0.6519**|**0.7521**|
> |w/o SSR|18.34|0.5742|0.6184|0.7196|
> |w/o AMG|20.08|0.5723|0.6047|0.7163|
> |w/o SSR & AMG|19.96|0.5642|0.6012|0.7058|
>
> These results confirm that both SSR and AMG contribute materially.
>
> **[Q3]: "Statistical analysis and validation under stronger degradations?"**
> **[A-Q3]:** Thanks for valuable suggestions. We added 95% confidence intervals and significance tests for the user study. All pairwise comparisons are statistically significant under a paired Wilcoxon signed-rank test ($p < 0.001$).
> |Methods|Rank Diff|95% CI|
> |---|---|---|
> |DP2O-SR|2.9|[2.2, 3.5]|
> |SeeSR|3.0|[2.6, 3.3]|
> |DreamClear|3.2|[2.8, 3.7]|
> |DiT4SR|2.7|[2.0, 3.4]|
>
> We incorporated additional evaluations with intensified degradations: see [A2] for RealLQ250 results and our response to Reviewer XsLC [A3] regarding the old films dataset.
>
> ---
> We hope our responses help address the concerns, and thanks again for the thoughtful feedback. We look forward to further discussion with you!
>
> [1] Wu et al., DP2O-SR, NeurIPS'25.
> [2] Wu et al., SeeSR, CVPR'24.
> [3] Ai et al., DreamClear, NeurIPS'24.
> [4] Duan et al., DiT4SR, ICCV'25.

---

> > ### Author Rebuttal · Reviewer_Gqoa · 2026-04-03
> >
> > Thank you for your detailed explanation, which address my concerns. I am positive about accepting this paper.

---

> > > ### Author Response · Authors · 2026-04-03
> > >
> > > Thank you very much for taking the time to read our rebuttal and for your very encouraging feedback! We are delighted to know that our response has fully addressed your concerns, and we are especially grateful for your comment that you are **"positive about accepting this paper"**.
> > >
> > > If you feel that our rebuttal has sufficiently resolved the main issues, we would be sincerely grateful if your final evaluation could reflect your updated view. In any case, we greatly appreciate your time, consideration, and support.

---

### Decision · Program_Chairs · 2026-04-30

**Decision:**

Accept (regular)

**Comment:**

This paper proposes the adversarial Sobolev alignment for faithful image super resolution. The paper originally received 2xWeakReject and 2xWeakAccept. The main concerns include less definitive theoretical grounding, robustness to unseen degradations, unclear advantages from the proposed Sobolev alignment itself, experimental protocol, claim about Riemannian geometry, etc. The authors have provided rebuttals and three reviewers mention that most of their concerns have been well addressed. Reviewer F8sV still questions about the overstatement of Riemannian geometry and the new ablation study. The authors have responded in detail in the discussion phase. Considering the rebuttal and discussions from all reviewers, ACs recommend weak acceptance for this paper. However, this paper currently presents certain unrigorous statements in its theoretical claims and the alignment between theory and practice remains imperfect. The authors are strongly suggested to revise the corresponding statements as appropriate and avoid over-claimed points as well as carefully revise the paper and incorporate newly conducted experiments according to the comments and discussions.